# Microstructural organization of human insula is linked to its macrofunctional circuitry and predicts cognitive control

Vinod Menon[1,2,3]*, Guillermo Gallardo[4,5], Mark A Pinsk[6], Van-Dang Nguyen[7], Jing-Rebecca Li[8], Weidong Cai[1], Demian Wassermann[9]*

[1]Department of Psychiatry and Behavioral Sciences, Stanford University School of Medicine, Stanford, Stanford, United States; [2]Department of Neurology and Neurological Sciences, Stanford University School of Medicine, Stanford, Stanford, United States; [3]Stanford Neurosciences Institute, Stanford University School of Medicine, Stanford, Stanford, United States; [4]Athena, Inria Sophia Antipolis, Université Côte d'Azur, Sophia Antipolis, France; [5]Department of Neuropsychology, Max Planck Institute for Human Cognitive and Brain Sciences, Leipzig, Germany; [6]Princeton Neuroscience Institute, Princeton University, Princeton, United States; [7]Department of Computational Science and Technology Royal Institute of Technology in Stockholm, Stockholm, Sweden; [8]Defi, Inria Saclay Île-de-France, École Polytechnique Université Paris Sud, Palaiseau, France; [9]Parietal, Inria Saclay Île-de-France, CEA Université Paris Sud, Palaiseau, France

*For correspondence:
menon@stanford.edu (VM);
demian.wassermann@inria.fr (DW)

Competing interests: The authors declare that no competing interests exist.

**Abstract** The human insular cortex is a heterogeneous brain structure which plays an integrative role in guiding behavior. The cytoarchitectonic organization of the human insula has been investigated over the last century using postmortem brains but there has been little progress in noninvasive in vivo mapping of its microstructure and large-scale functional circuitry. Quantitative modeling of multi-shell diffusion MRI data from 413 participants revealed that human insula microstructure differs significantly across subdivisions that serve distinct cognitive and affective functions. Insular microstructural organization was mirrored in its functionally interconnected circuits with the anterior cingulate cortex that anchors the salience network, a system important for adaptive switching of cognitive control systems. Furthermore, insular microstructural features, confirmed in Macaca mulatta, were linked to behavior and predicted individual differences in cognitive control ability. Our findings open new possibilities for probing psychiatric and neurological disorders impacted by insular cortex dysfunction, including autism, schizophrenia, and fronto-temporal dementia.

## Introduction

The human insular cortex plays a critical role in identifying salient sensory, affective, and cognitive cues for guiding attention and behavior (*Nieuwenhuys, 2012*; *Menon and Uddin, 2010*; *Uddin, 2015*; *Craig, 2009*). Critically, it is also one of the most widely activated brain regions in all of human neuroimaging research (*Cai et al., 2014*; *Swick et al., 2011*; *Levy and Wagner, 2011*). Dysfunction of the human insula and its interconnected regions are now thought to be core features of many psychiatric and neurological disorders (*Goodkind et al., 2015*; *Namkung et al., 2017*). However, little is known about the normative microstructural organization of the human insular cortex and its relation to behavior. Precise quantitative in vivo characterization of the microstructural

organization of the human insular cortex and links to its functional circuitry are critical for understanding its function and role in development and psychopathology.

The insular cortex is a structurally heterogeneous brain region with a distinct cytoarchitectonic profile characterized by less differentiated cortical layers (*Nieuwenhuys, 2012*; *Mesulam and Mufson, 1982a*; *Morel et al., 2013*; *Evrard, 2019*). Investigations of the microstructural features of the human insular cortex have been based solely on postmortem brains with small samples and limited behavioral characterizations. Insular cytoarchitectonic organization has been investigated using histological techniques over the last century (*Allman et al., 2010*; *Brodmann, 1909*; *Von Economo et al., 2008*; *Mesulam and Mufson, 1985*; *Augustine, 1996*). Stereological analyses of the insula have identified a cellular cortical architecture which differs considerably from the 6-layer granular architecture seen in most cortical areas (*Mesulam and Mufson, 1982a*; *Mesulam and Mufson, 1982b*; *Augustine, 1985*). In a seminal study, Mesulam and Mufson (*Mesulam and Mufson, 1982b*) proposed the concept of 'granularity', based on the presence of an inner granular layer, as a key feature for identifying the anatomical subdivisions of the insular cortex. In the ensuing years, several histological studies have focused on demarcating the microstructural properties of insular subdivisions but no consensus has yet emerged about their precise boundaries (*Morel et al., 2013*; *Allman et al., 2010*; *Kurth et al., 2010*). Thus far, investigations of the distinct structural features of the human insular cortex have been based solely on postmortem brains and have not been amenable to characterization using non-invasive brain imaging techniques. Quantitative modeling of non-invasive in vivo brain imaging is therefore critically needed to address this major gap.

Despite the lack of consistency and precision across previous histological studies, some general patterns have emerged regarding the general cytoarchitectonic organization of the insula. The ventral anterior insula has an agranular structure characterized by undifferentiated layers II/III, distinct from the fully developed granular cortex with a canonical 6-layer architecture. The dorsal anterior insula and most central territories of the insula display an intermediate dysgranular profile (*Mesulam and Mufson, 1982a*; *Morel et al., 2013*; *Mesulam and Mufson, 1985*). In contrast, large sections of the posterior insula show a canonical granular structure (*Kurth et al., 2010*). Another unique aspect of the neuronal organization of the human insula is the presence of von Economo neurons in the anterior aspects of the insula (*Namkung et al., 2017*). These neurons differ from the typical pyramidal neurons by virtue of their large spindle shape and thick basal and apical dendrites which allow for speeded communication (*Seeley et al., 2012*; *von Economo, 1926*). The dysgranular organization of the human insula, together with the presence of specialized neurons, have been hypothesized to contribute to its crucial role in goal-directed behaviors and emotional regulation, through rapid processing of attentional, cognitive, interoceptive, emotional, and autonomic signals (*Craig, 2009*; *Seeley et al., 2007*; *Critchley et al., 2004*). However, the lack of tools for assessing morphological variations in the insula in vivo and its relation to behavior has limited our understanding of the microstructural organization of the insula in health and disease.

The insular cortex is also functionally heterogeneous and integrates signals across its cognitive and affective subdivisions to support adaptive behavior. The anterior aspects of the insula are important for detection of salient external stimuli and for mediating goal-directed cognitive control while the posterior aspects are important for integrating autonomic and interoceptive signals (*Craig, 2009*). This functional organization is supported by a distinct pattern of long-range connections: the dorsal aspects of the anterior insula (AI) are more strongly connected to brain areas important for cognitive control, most notably the dorsal anterior cingulate cortex (ACC), while the ventral anterior has stronger links with subcortical and limbic regions important for emotion, reward and homeostatic regulation, including the amygdala, ventral striatum and hypothalamus (*Menon and Uddin, 2010*; *Craig, 2009*; *Dosenbach et al., 2007*; *Chikama et al., 1997*). Consistent with these reports, meta-analysis of task-related coactivation patterns in human neuroimaging studies point to distinct functional networks associated with insular subdivisions (*Chang et al., 2013*). Critically, the anterior insula and the dorsal ACC anchor the salience network (SN), a tightly coupled network that is among the most widely co-activated set of brain regions in all of human neuroimaging research (*Cai et al., 2014*; *Swick et al., 2011*; *Levy and Wagner, 2011*). Remarkably, besides the insula, the only other brain region where von Economo neurons are known to be strongly expressed is the ACC (*Allman et al., 2010*; *Nimchinsky et al., 1999*). However, the link between the microstructural organization of the insula and its functional circuit properties is currently not known. Furthermore, it is unclear whether these regional functional circuit properties are mirrored in the long-range

connectivity of the insula in the context of the large-scale organization of AI-ACC circuits that anchor the SN. To the best of our knowledge there have been no histological investigations of the correspondence between cytoarchitecture features of the AI and ACC.

Given the challenges inherent in obtaining histological data from postmortem brains, the bulk of current research on the human insula has focused on its functional organization both in relation to regional activation by specific cognitive and affective tasks and its distinct patterns of functional connectivity with other brain regions (*Cai et al., 2014*; *Chang et al., 2013*; *Deen et al., 2011*; *Ryali et al., 2015*; *Jakab et al., 2012*; *Cauda et al., 2012*; *Kelly et al., 2012*). Crucially, non-invasive voxel-wise quantitative mapping of functional brain connectivity has allowed researchers to identify distinct subdivisions within the insular cortex (*Deen et al., 2011*; *Ryali et al., 2015*; *Faillenot et al., 2017*). Based on their unique fingerprints of connectivity, previous research has identified three distinct functionally subdivisions in human insular cortex: the dorsal anterior (dAI), ventral anterior (vAI), and posterior insula (PI) (*Namkung et al., 2017*; *Deen et al., 2011*; *Glasser et al., 2016*). These subdivisions also show distinct patterns of intrinsic functional connectivity within the SN, with the dAI being more tightly linked to the ACC node of the SN (*Deen et al., 2011*). Whether functionally-defined regions of the insula and its functional circuits associated with the SN have distinct microstructural features is currently not known, and addressing this link has the potential to contribute to a deeper understanding of structure-function relations in the human brain.

Noninvasive in vivo investigation of tissue microstructure is a key application of diffusion MRI (dMRI), but few studies have tapped its potential due to limitations of most previous dMRI techniques (*Calamante et al., 2018*). A key discovery that now permits more precise microstructural analysis of human gray matter arose from the work by *Latour et al. (1994)* demonstrating the feasibility of analyzing cellular size in biological tissue with time-dependent diffusion MR. In subsequent studies, dMRI was used to determine sensitivity of dMRI signals in gray matter tissue (*Basser and Pierpaoli, 2011*; *Thornton et al., 1997*). More recently, dMRI has been used to characterize normal human brain maturation (*Mukherjee et al., 2002*) and cortical microstructure in preterm infants (*Ball et al., 2013*). Building on these studies, multi-shell models have recently been used to demonstrate regional variability in cortical microstructure (*Calamante et al., 2018*; *Fukutomi et al., 2018*), as well as sensitivity of multi-shell dMRI for probing fine-scale and region-specific microstructure using high angular and spatial resolution diffusion MRI (*Aggarwal et al., 2015*; *McNab et al., 2013*).

Here we leverage recent advances in multi-shell dMRI acquisition protocols and recent signal reconstruction techniques (*Özarslan et al., 2013*; *Fick et al., 2016*) to determine the microstructural features of insular cortex using the normalized Return to Origin Probability (RTOP) density index (*Mitra et al., 1995*; *Schwartz et al., 1997*). RTOP measures water diffusion in neural tissue (*Mitra et al., 1995*), with higher values reflecting more restricted diffusion. RTOP has been shown to be inversely correlated with the average volume of pores restricting diffusion in microbeads (*Mitra et al., 1995*; *Schwartz et al., 1997*). In neural tissue, spinal cord maturation has been shown to increase RTOP (*Assaf et al., 2000*). Recently, RTOP has been shown to be more sensitive to cellular organization than other more commonly used diffusion MRI measures, such as mean and radial diffusivity (*Avram et al., 2016*). Based on these observations, we hypothesized that RTOP in grey matter would be sensitive to the cytoarchitectonic and neuronal organization of the insular cortex.

The first aspect of our study involved leveraging a large (N = 413) cohort of adult participants from the Human Connectome Project (HCP) (*Glasser et al., 2013*). We evaluated RTOP across three major functional subdivisions of the human insula and demonstrate that they have microstructurally distinct profiles. Based on histological observations, we predicted smaller RTOP values in its dorsal and anterior subdivisions, consistent with their agranular and dysgranular organization, compared to the posterior insula and its granular organization. Next, because of the paucity of insula-wide histological data in humans, we used dMRI data from two macaques to examine whether RTOP measures are sensitive to known cytoarchitectural features of the primate insula (*Brockhaus, 1940*; *Rose, 1928*). We then examined whether the microstructural distinctions seen in the insular cortex are mirrored in the ACC node of the salience network, and demonstrate a strong link between functional circuits linking the insula and ACC and their microstructural organization in human brain. Crucially, we examine whether variation in microstructural features of the insula are related to behavior and individual differences in cognitive control ability. Stability, cross-validation and replication

analyses are used to demonstrate the robustness of our findings. *Figure 1* provides an overview of the data analysis pipeline.

## Results

### Insula microstructure across its functionally defined subdivisions

We characterized insula microstructure across three major functional subdivisions of the insula (*Deen et al., 2011*; *Glasser et al., 2016*): dAI, vAI, and PI. RTOP (*Mitra et al., 1995*) measures in each subdivision were computed by averaging across all mesh-vertices in each subdivision (see Appendix 1 for details of the mathematical formulation and computations).

We first examined insula subdivisions using the functional parcellation derived by Deen and colleagues (*Deen et al., 2011*; *Figure 2A*). To determine whether RTOP values differed among the three insular subdivisions, we conducted an ANOVA with factors subdivision (vAI, dAI and PI) and hemisphere (left vs. right). We found a significant interaction between hemisphere and subdivision (ANOVA, $F_{(2,824)} = 33.7$, p<8.2E-15) and significant main effects of subdivision ANOVA, ($F_{(2,824)} = 1102$, p<2E-16) and hemisphere (ANOVA, $F_{(1,412)} = 50.2$, p<6.1E-12). Post-hoc *t*-tests further revealed a gradient of RTOP values vAI <dAI < PI in both right and left hemisphere (paired t-tests, all *ps* <2.0E-11, except p=0.037 in the case of right vAI <dAI), with vAI and PI having significantly smaller values in the right, compared to the left, hemisphere (paired t-tests, *ps* <1.6E-15) (*Figure 2B*).

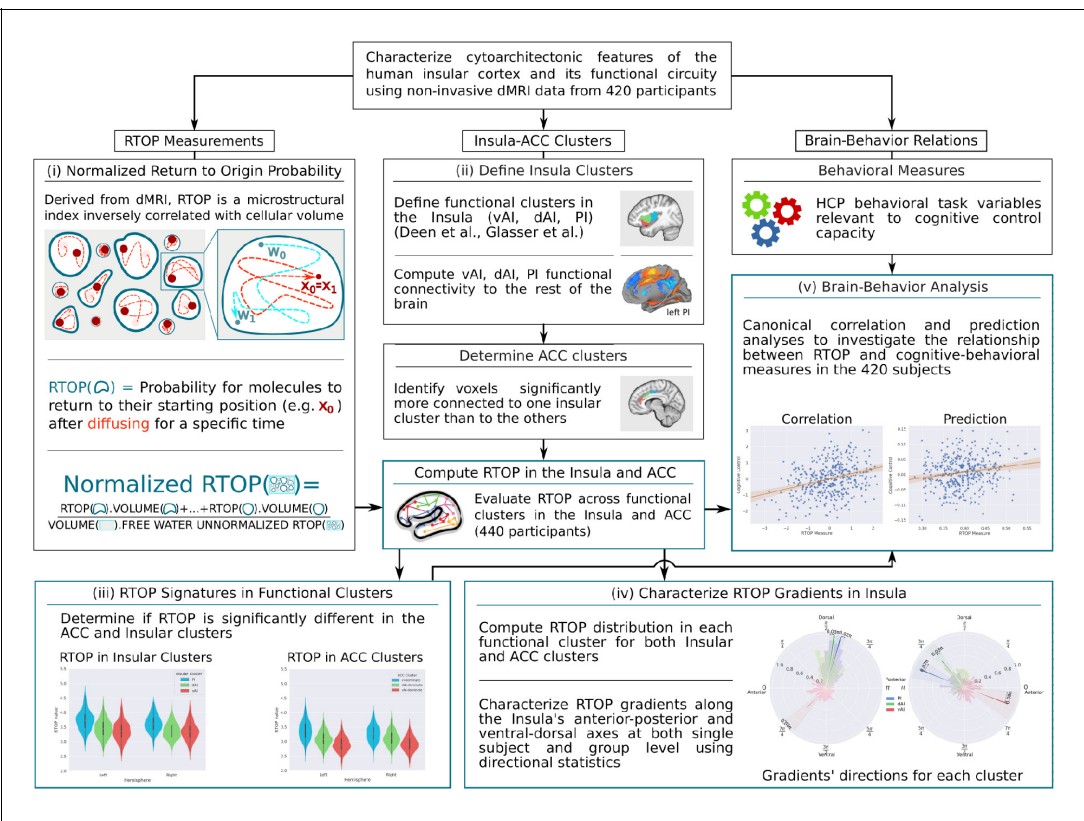

**Figure 1.** Flow chart illustrating data analysis pipeline. (**A**) Main components of human insula microstructure analysis using diffusion magnetic resonance imaging (dMRI) data from 413 Human Connectome Project (HCP) participants. Key steps include: (i) measurement of microstructure features based on Return to Origin Probability (RTOP), the ratio between the probability of molecules returning to their starting position in biological tissue versus free diffusion, (ii) demarcation of functional subdivisions in insula and its interconnected anterior cingulate cortex (ACC) subdivisions, which together anchor the salience network, (iii) computation of microstructure features of the insula within its functional subdivisions and its interconnected ACC subdivisions, (iv) computation of microstructural gradients along the anterior-posterior and dorsal-ventral axes of the insula, and (v) analysis of relation between insula microstructural organization and cognitive control abilities.

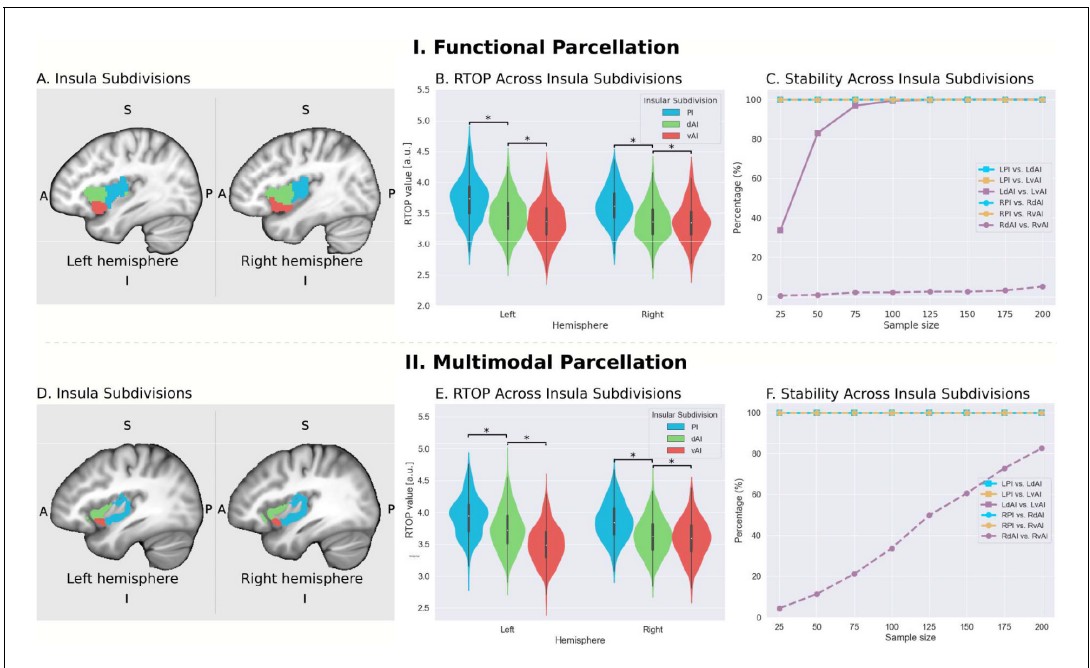

**Figure 2.** Microstructural properties of functional insular cortex subdivisions. (**A**) Functional subdivisions spanning the posterior insula (PI), dorsal anterior insula (dAI) and ventral anterior insula (vAI) (*Deen et al., 2011*). (**B**) RTOP differs significantly between PI, dAI and vAI in the left hemisphere and between PI and dAI or vAI in the right hemisphere (p<0.001, Bonferroni corrected). The right vAI has the smallest RTOP value among all the subdivisions (all *ps* < 0.001, except p=0.036 for right vAI < dAI; Bonferroni corrected). (**C**) Stability of findings as a function of sample size. A sample size of N = 25 was sufficient to achieve a stable differentiation (p<0.01) between PI and vAI in both hemispheres, while differentiating the vAI and dAI required a larger sample size. a PI: posterior insula; dAI: dorsal anterior insula; vAI: ventral anterior insula. (**D–F**) Replication with an independent multimodal parcellation using HCP data (*Glasser et al., 2016*).

The online version of this article includes the following source data and figure supplement(s) for figure 2:

**Source data 1.** Data for *Figure 2* panel B.
**Source data 2.** Data for *Figure 2* panel E.
**Figure supplement 1.** RTOP measurements validate the cytoarchitectonic organization of the macaque insular cortex in vivo.

Next, we examined the stability of these findings as a function of sample size. We found that a sample size of N = 25 was sufficient to achieve a stable differentiation (p<0.01) between PI and vAI in both hemispheres, while differentiating the vAI and dAI required a larger sample size of N = 100 (*Figure 2C*).

We then conducted a replication analysis using a multimodal parcellation of the insula from a large sample of HCP participants (*Glasser et al., 2016*; *Figure 2D*). Here again, we found a significant interaction between hemisphere and subdivision (ANOVA, F(2,824) = 207.3, p<2E-16) and a significant main effect of subdivision (ANOVA, F(2,824) = 11265, p<2E-16) and hemisphere (ANOVA, F(1,412) = 10.9, p<0.001). Post-hoc *t*-tests further revealed a gradient of RTOP values vAI <dAI < PI in both right and left hemispheres (paired t-test, all *ps* <3.2E-06), with vAI and PI having significantly smaller values in the right hemisphere and vAI having significantly smaller values in the left hemisphere (paired t-test, *ps* <4E-13) (*Figure 2E*).

Stability analysis revealed that a sample size of N = 25 was sufficient to achieve a stable differentiation (p<0.01) between PI and vAI and between PI and dAI in both hemispheres and between dAI and vAI in the left hemisphere (*Figure 2F*).

Our findings provide robust and replicable evidence for distinct microstructural variations across these functional subdivisions of the insula, and suggest a close correspondence between the microstructural and functional organization of the insula.

## Gradient analysis of insula microstructural features along the anterior-posterior and dorsal-ventral axes

To further delineate the microstructural organization of the human insula, we conducted a detailed profile analysis and used directional statistics to characterize gradients in RTOP along its anterior-posterior and ventral-dorsal axes (*Figure 3*). *Figure 3A* shows isolevels of RTOP values in the insula at a group level. We found a low dispersion of RTOP gradients in both the left (0.36π ±0.09 π, SEM = 0.005) and right (0.36π ±0.09 π, SEM = 0.005) hemispheres. We then assessed the significance of this finding using the Rayleigh test, against the null hypothesis of a uniform directional distribution (*Mardia, 2000*). We found strong evidence for both an anterior-to-posterior and ventral-to-dorsal gradient in the left insula ($c_i$ = 0.02π, Rayleigh statistic = 724, p<1.0e-10, N = 413, df = 3) and the right insula ($c_i$ = 0.04π, Rayleigh statistic = 270., p<1.0e-10, N = 413, df = 3) (*Figure 3B–C*, *Supplementary file 1*-Supplementary Table 1). The confidence interval showed a low dispersion, indicating a consistent pattern of gradients across participants (*Supplementary file 1*-Supplementary Table 1). The lowest values of RTOP were localized to the vAI, convergent with findings from the analysis of the three functional subdivisions described above. Finally, analysis of microstructure isolines further revealed gradient 'fingers' from the vAI extending along a ventral-dorsal axis (*Figure 4*). We repeated these analyses using the multi-modal parcellation. Here again we found low dispersion in RTOP gradients in both left (0.16π ±0.07π, SEM = 0.01) and right hemisphere (0.23 ± 0.09π, SEM = 0.01). We again found evidence for an anterior-posterior and ventral-dorsal gradients spanning from vAI extending along a ventral-dorsal axis through PI (*Figures 3E–F* and *4B*, *Supplementary file 1*-Supplementary Table 2). These results point to a consistent and reliable pattern of gradients and microstructural organization of the human insula along its anterior-posterior and dorsal-ventral axes.

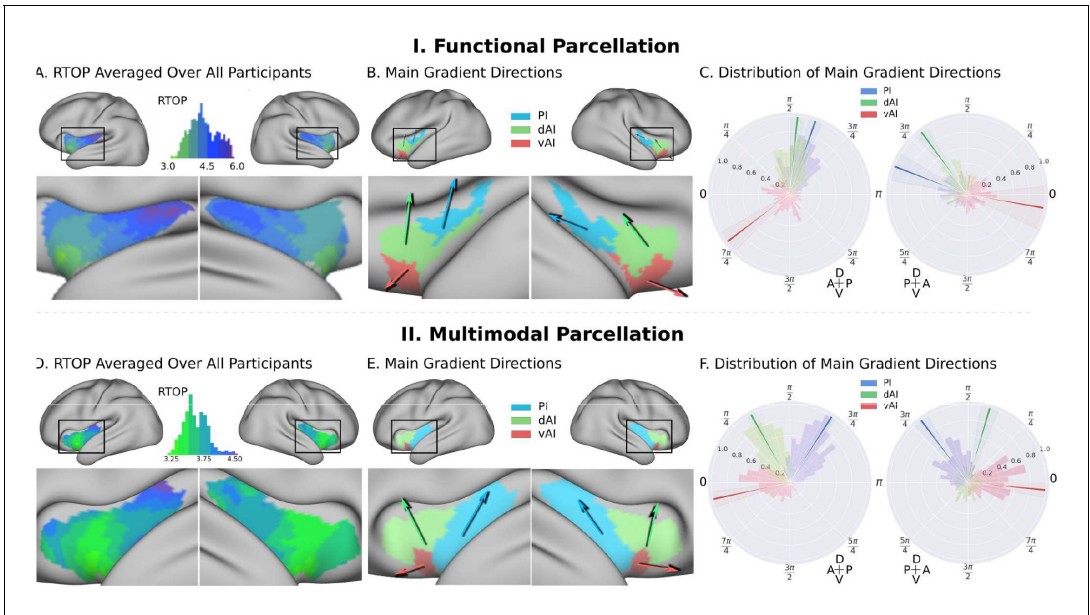

**Figure 3.** Insula microstructure gradients along its anterior-posterior and dorsal-ventral axes. (A) RTOP averaged over all participants (N = 413) illustrates inhomogeneity in insula microstructure with a ventral anterior insula peak, and gradients along the anterior-posterior and dorsal-ventral gradients axes. Larger RTOP indicates smaller average compartments. There is a prominent gradient from the insular pole towards the posterior insular section. Note right hemisphere dominance. (B) Main gradient directions, computed using Rayleigh directional statistics in each functionally defined subdivision. The main directions show an anterior-to-posterior and inferior-to-superior RTOP organization in the left insular cortex and an anterior-to-posterior organization in the right insular cortex. The polar plots show the distribution of main gradient directions in each functional subdivision. (C) Gradient direction histograms. The mean direction is represented with solid lines on top of the distribution histogram; the shaded region represents the 95% confidence interval. For detailed statistics (Table S1). (D–F) Replication with an independent multimodal parcellation using HCP data (*Glasser et al., 2016*).

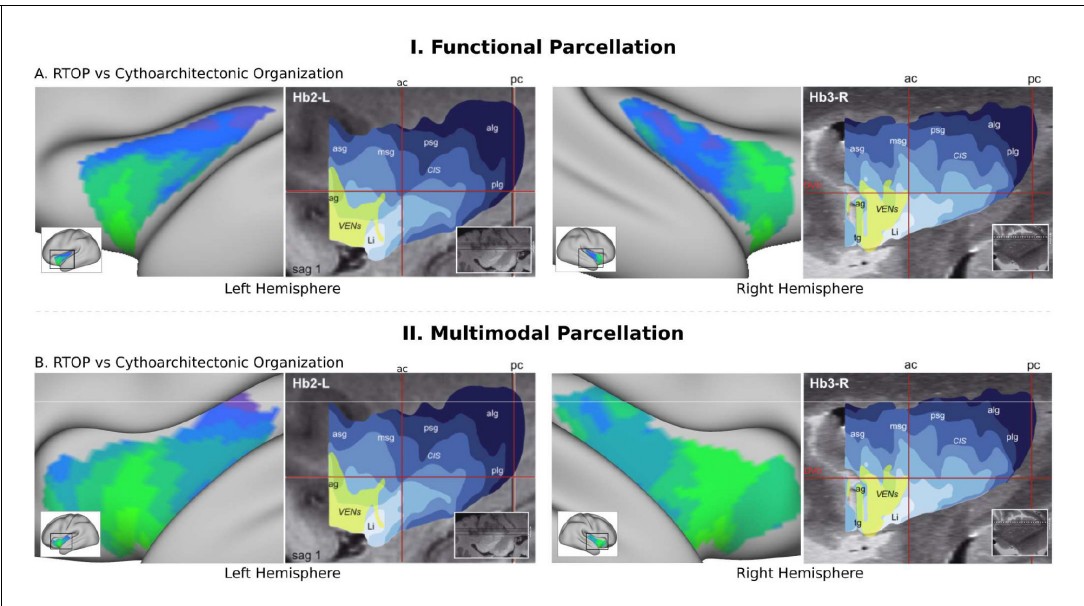

**Figure 4.** RTOP isocontours reflect insula cytoarchitectonic organization. Isocontours of the population-average RTOP (left) are closely aligned to cytoarchitectonic organization of the insula and VEN expression from studies of post-mortem brains (right, based on *Morel et al., 2013*). (B) Replication with an independent.

## RTOP captures known microstructural features of the insular cortex in macaques

We next used dMRI data from two macaques to examine whether RTOP measures are sensitive to known cytoarchitectural features of the non-human primate insula. Histological studies have demonstrated that the macaque insula can be demarcated into agranular, dysgranular and granular subdivisions based on their unique cytoarchitectural properties (*Brockhaus, 1940*; *Rose, 1928*; *Evrard et al., 2012*; *Evrard et al., 2014*; *Nieuwenhuys, 2012*). To investigate whether RTOP measures are sensitive to known cytoarchitectural features of the primate insula, we acquired dMRI data from two macaque monkeys using protocols similar to the HCP (Appendix 1), and examined RTOP values in the three known cytoarchitectonic subdivisions of the primate insula. We found that RTOP captures known microstructural features of the insular cortex in macaques (Appendix 2; *Figure 2—figure supplement 1*, *Supplementary file 1*-Supplementary Table 3), providing convergent validation of our in vivo human RTOP findings.

## Microstructural features in the insula and ACC nodes of the salience network

The insular cortex and ACC are the two major cortical nodes of the SN (*Menon and Uddin, 2010*; *Seeley et al., 2007*). Individual functional subdivisions of the insula have preferential connections to different subdivisions of the ACC (*Chang et al., 2013*; *Deen et al., 2011*; *Cauda et al., 2012*; *Taylor et al., 2009*), but it is not known whether they share similar microstructural features. To address this gap, we first conducted a seed-based whole-brain functional connectivity analysis, where seeds are the three functional subdivisions of the insular cortex in each hemisphere (*Deen et al., 2011*).

We first examined the insular functional parcellation (*Deen et al., 2011*). Our analysis revealed that the three insular subdivisions had distinct functional connectivity patterns with the ACC (*Figure 5A*). Specifically, the vAI showed stronger connection with the most anterior and ventral ACC (denoted as ACC-vAI), the dAI showed stronger connection with the middle and dorsal ACC (ACC-dAI), and the PI showed stronger connections with the posterior ACC (ACC-PI) (*Figure 6A*).

Next, we created functional subdivisions within the ACC based on their differential connectivity patterns with the dAI, vAI and PI. First, we examined functional connectivity differences between each pair of seeds in each hemisphere (e.g. left PI >left dAI, thresholded, p<0.01, FDR corrected).

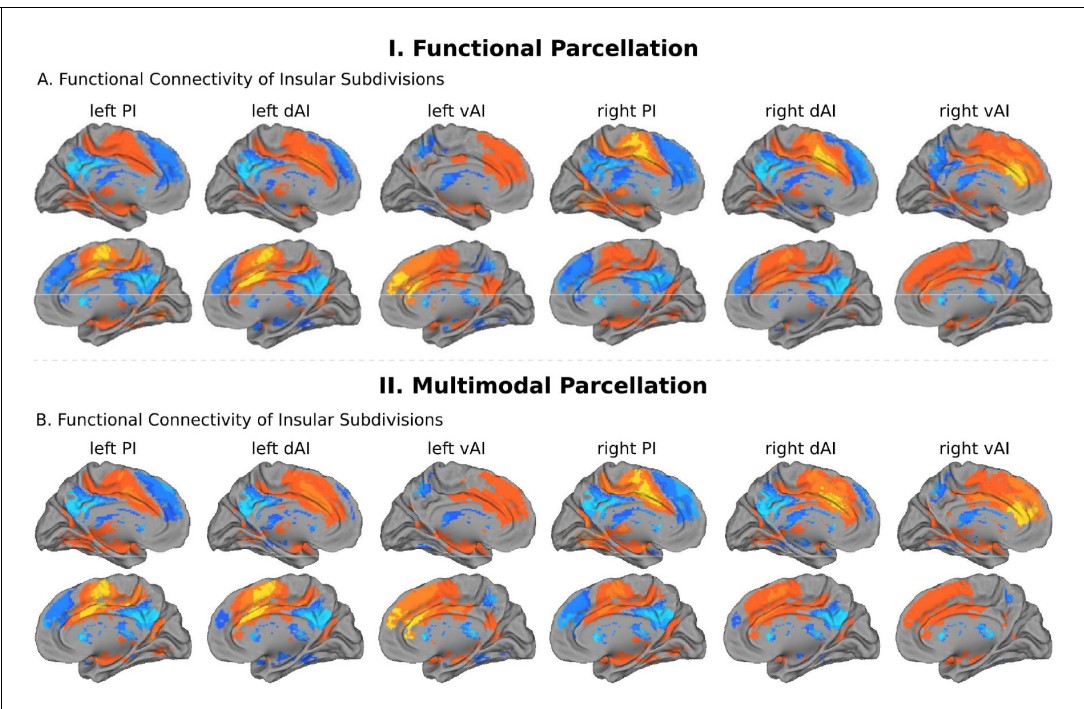

**Figure 5.** Intrinsic functional connectivity of insula subdivisions dAI, vAI and PI. (**A**) Whole-brain resting-state functional connectivity analysis revealed distinct functional connectivity patterns of the insula subdivisions (*Deen et al., 2011*), bilateral posterior insula (PI), dorsal anterior insula (dAI) and ventral anterior insula (vAI) (all *ps* < 0.001, FDR corrected). (**B**) Replication with an independent multimodal parcellation using HCP data (*Glasser et al., 2016*).

We then applied a logical AND operation to identify voxels surviving the two binarized maps that indicated the voxel's connectivity to one insula subdivision is significantly greater than the others (e.g. left PI >left dAI and left PI >left vAI). Finally, the resulting binarized maps were overlapped with an ACC mask (*Figure 7A*).

To determine whether RTOP differs across the ACC functional subdivisions linked to individual insular divisions we conducted an ANOVA with factors subdivision (ACC-vAI, ACC-dAI and ACC-PI) and hemisphere (left vs. right). We found a significant interaction between subdivision and hemisphere (ANOVA, $F_{(2,824)}$ = 1014, p=2E-16), and significant main effects of subdivision (ANOVA, $F_{(2,824)}$ = 1151, p=2E-16) and hemisphere (ANOVA, $F_{(1,412)}$ = 330, p=2E-16). Post-hoc paired *t*-tests revealed significant differences in RTOP: ACC-vAI < ACC-dAI < ACC-PI in both hemispheres (paired t-tests, all *ps* < 2E-16) (*Figure 7B*). Stability analysis demonstrated that these differences were highly reliable (paired t-tests, p<0.01) for samples sizes > N = 25 (*Figure 7C*).

We then conducted a replication analysis using a multimodal HCP insular parcellation (*Glasser et al., 2016*). Once again, the three insular subdivisions had distinct functional connectivity patterns with the ACC (*Figure 5B*). Specifically, the vAI showed stronger connection with the most anterior and ventral ACC (ACC-vAI), the dAI showed stronger connection with the middle and dorsal ACC (ACC-dAI), and the PI showed stronger connections with the posterior ACC (ACC-PI) (*Figure 6B*). We found a significant interaction between ACC subdivisions (*Figure 7D*) and hemisphere (ANOVA, $F_{(2,824)}$ = 245.7, p=2E-16), and significant main effects of subdivision (ANOVA, $F_{(2,824)}$ = 343, p=2E-16) and hemisphere (ANOVA, $F_{(1,412)}$ = 248.9, p=2E-16) (*Figure 7E*). Post-hoc paired *t*-tests revealed significant differences in RTOP: ACC-vAI < ACC-dAI < ACC-PI in both hemispheres (paired t-tests, all *ps* < 6E-11) (*Figure 7E*). Stability analysis demonstrated that the differences between PI and vAI and between dAI and vAI in both hemisphere and between PI and dAI in the right hemisphere were highly reliable (paired t-tests, p<0.01) for samples sizes > N = 50 (*Figure 7F*).

The three ACC subdivisions overlapped with distinct areas demarcated by the HCP multimodal atlas (*Glasser et al., 2016*): the ACC-vAI overlapped with ACC area p24, ACC-dAI overlapped with

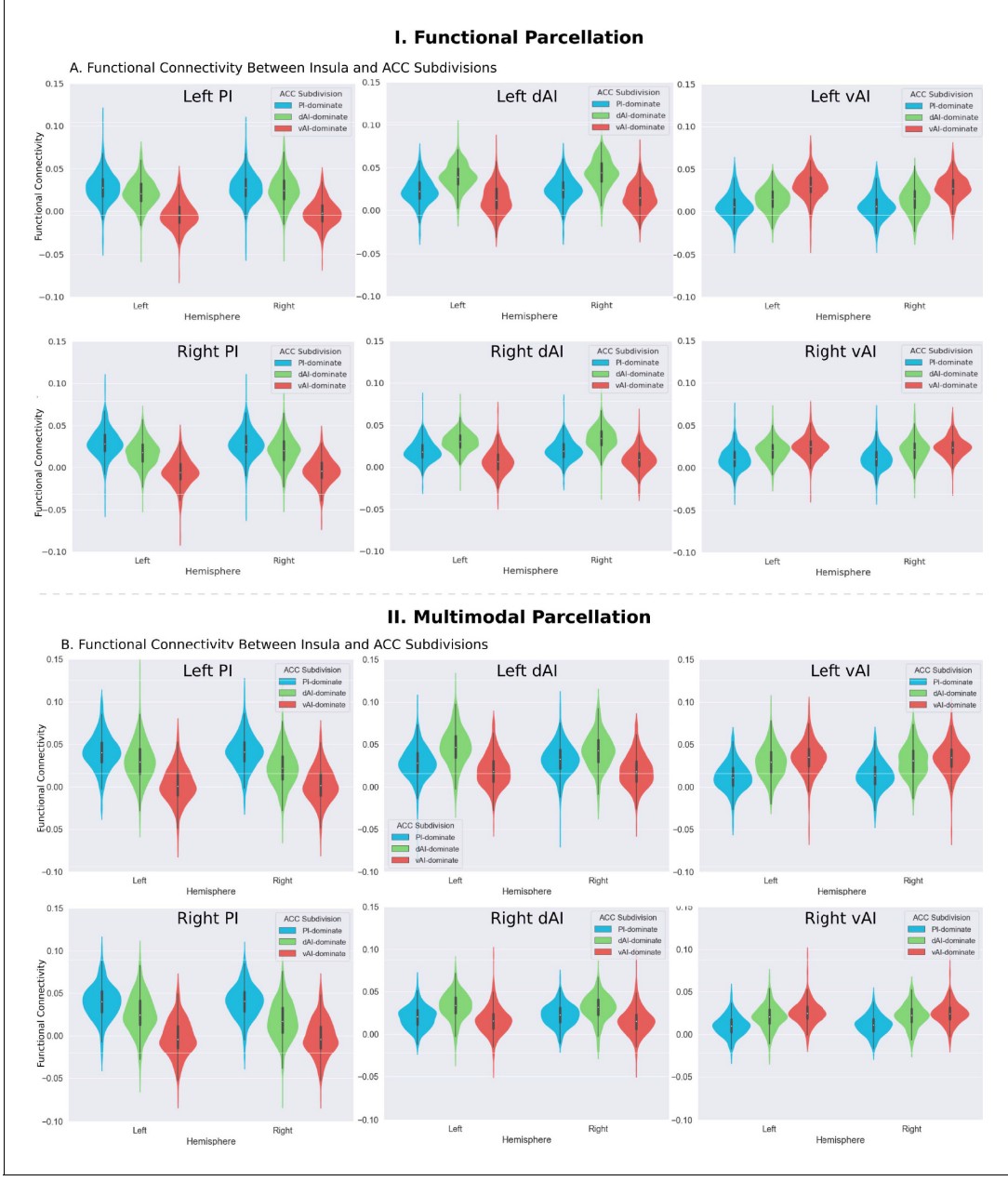

**Figure 6.** Functional connectivity between insula and ACC subdivisions. (**A**) RTOP differs across ACC functional subdivisions differentially linked to the three insular subdivisions (**Deen et al., 2011**). (**B**) Replication with an independent multimodal parcellation using HCP data (**Glasser et al., 2016**). The online version of this article includes the following source data for figure 6:

**Source data 1.** Data for *Figure 6* panel A.
**Source data 2.** Data for *Figure 6* panel B.

cingulate area ap24pr, and ACC-PI overlapped with ACC area p24pr, thus demonstrating distinct patterns of connectivity linking insula and ACC subdivisions.

These results demonstrate that the microstructural organization of the ACC mirrors the microstructural organization of the three insula subdivisions with which it is differentially connected.

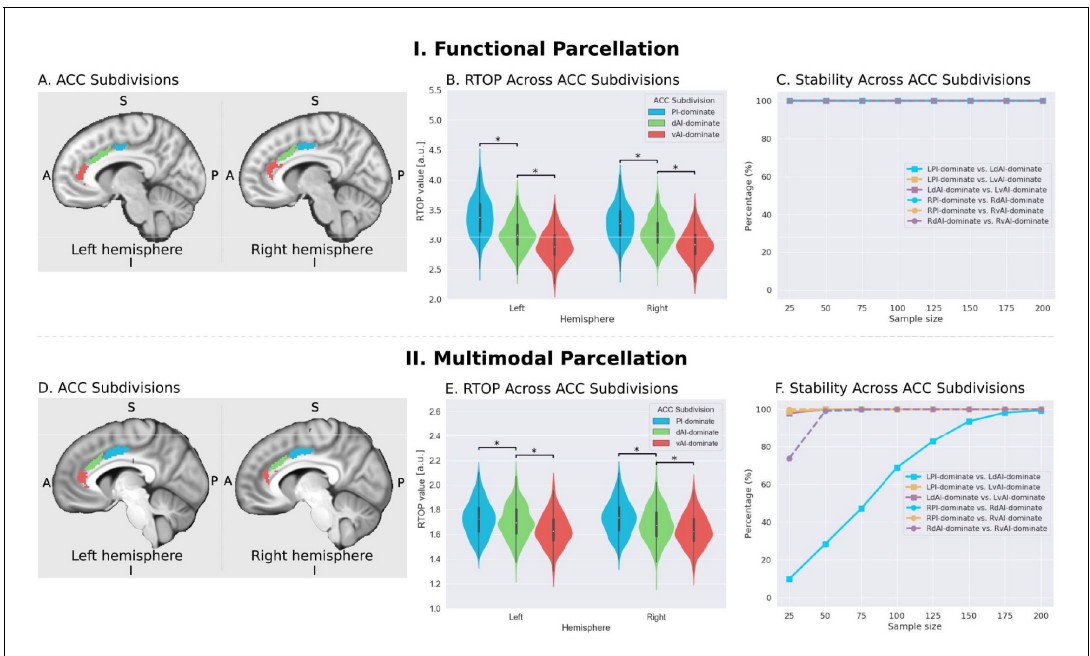

**Figure 7.** Microstructural properties of anterior cingulate cortex subdivisions mirror profiles in corresponding insula functional subdivisions. (**A**) Illustration of ACC subdivisions. Each ACC subdivision preferentially connects to one of the three insular subdivisions defined using an independent functional parcellation (*Deen et al., 2011*). ACC subdivisions showed significantly greater functional connectivity with one insula subdivision over others (e.g. right PI > right dAI) and (right PI > right vAI) (all *ps* < 0.01, FDR corrected). (**B**) RTOP values were significantly different among the three ACC subdivisions in each hemisphere (p<0.001, Bonferroni corrected). The ACC subdivision differentially connected to vAI has smaller RTOP values than the other subdivisions (all *ps* < 0.001, Bonferroni corrected). (**C**) RTOP differences among three ACC subdivisions were robust and reliable at sample sizes of N = 25 or more. PI: posterior insula; dAI: dorsal anterior insula; vAI: ventral anterior insula. (**D–E**) Replication with an independent multimodal parcellation using HCP data (*Glasser et al., 2016*).

The online version of this article includes the following source data for figure 7:

**Source data 1.** Data for *Figure 7* panel B.
**Source data 2.** Data for *Figure 7* panel E.

## Insula microstructure and relation to cognitive control ability

The human insula plays an important role in detection of salient external stimuli and in mediating goal-directed cognitive control (*Craig, 2009*). We investigated the relationship between microstructural properties of the insula and cognitive control ability, using canonical correlation analysis (CCA) with cross-validation and prediction analysis. Mean RTOP values from the six insular subdivisions, three in each hemisphere, were used to predict cognitive-behavioral measures associated with processing speed, working memory, response inhibition and cognitive flexibility.

We first used the insular functional parcellation (*Deen et al., 2011*). We found a significant relation between insula RTOP values and individuals cognitive control abilities (Canonical correlation, Pillai's trace = 0.21, p<0.05). The canonical weights of the 1st latent variable in microstructural measures were significantly correlated with the canonical weights of the 1st latent variable in behavioral measures (*Pearson*'s correlation, r = 0.32, p<0.001, *Cohen's d* = 0.67, *Figure 8A*; *Supplementary file 1*-Supplementary Table 4). A leave-one-out cross-validation procedure further revealed that, based on microstructural properties of the insula, our CCA model could predict cognitive control ability on unseen data (*Pearson*'s correlation, r = 0.19, p<0.001, *Cohen's d* = 0.39, *Figure 8B*), with the strongest predictive weights in the right dAI.

Finally, we conducted a replication analysis using the multimodal HCP parcellation (*Glasser et al., 2016*). We found a marginally significant relationship between insula RTOP values and individuals' cognitive control abilities (Canonical correlation, Pillai's trace = 0.20, p=0.08). The canonical weights of the 1st latent variable in microstructural measures were significantly correlated with the canonical weights of the 1st latent variable in behavioral measures (*Pearson*'s correlation, r = 0.31, p<0.001,

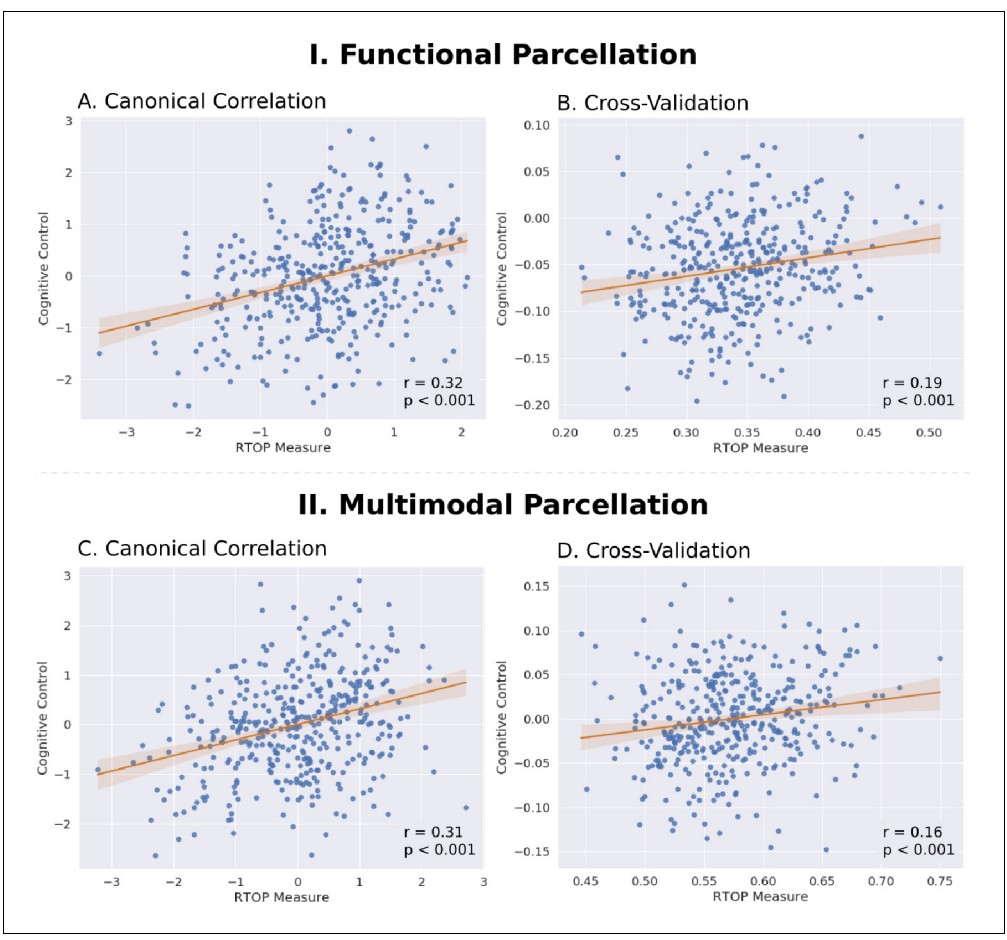

**Figure 8.** Insula microstructural features predict cognitive control abilities. (**A**) Canonical correlation analysis (CCA) revealed a significant relationship between mean RTOP in each insular subdivision (**Deen et al., 2011**) and cognitive control measures. CCA weights of RTOP measures were significantly correlated with CCA weights of cognitive control measures in CCA Axis 1. (**B**) Cross-validation analysis revealed that CCA-derived weights of RTOP predicted cognitive control measures on unseen data. (**C–D**) Replication with an independent multimodal parcellation using HCP data (**Glasser et al., 2016**).

The online version of this article includes the following source data for figure 8:

**Source data 1.** Data for **Figure 8** panel A.
**Source data 2.** Data for **Figure 8** panel B.
**Source data 3.** Data for **Figure 8** panel C.
**Source data 4.** Data for **Figure 8** panel D.

Cohen's d = 0.65, **Figure 8C**; **Supplementary file 1**-Supplementary Table 5). A leave-one-out cross-validation procedure further revealed that, based on microstructural properties of the insula, our CCA model could predict cognitive control ability on unseen data (*Pearson*'s correlation, r = 0.17, p<0.001, *Cohen's d* = 0.35, **Figure 8D**), with the strongest predictive weights in the right dAI.

## Discussion

We leveraged recent advances in diffusion weighted imaging of gray matter and large high-quality samples from the HCP to investigate the microstructural properties of the insular cortex and its macrofunctional circuits associated with the salience network. Novel quantitative modeling of in vivo dMRI and fMRI signals allowed us to probe links between the functional subdivisions of the insula and its microstructure properties, overcoming limitations of postmortem studies. We found that the human insular cortex is characterized by a systematic profile of microstructural variation across its

distinct functional subdivisions. Beyond the boundaries of the functionally defined subdivisions, our analysis revealed gradients along the anterior-posterior axis as well as dorsal-ventral axis that are consistent with known cytoarchitectonic differences in the insula derived from studies of post-mortem human brains (*Morel et al., 2013*; *Ding et al., 2016*). Notably, we observed significant hemispheric differences with the right hemisphere showing lower RTOP values than the left hemisphere. Critically, the microstructural organization of the insula was mirrored in the dorsal ACC, which together forms the backbone of the salience network, a system important for rapid orienting of attention and adaptive switching of cognitive control systems (*Sridharan et al., 2008*). Remarkably, microstructural properties of the insula were correlated with cognitive control abilities, and furthermore predicted these abilities in a left-out sample. Crucially, we replicated all the major findings using two independent insular parcellations. Our findings provide novel insights into human insular cortex architecture and its contribution to human cognition.

## Insula microstructure differs across functionally defined subdivisions

We used multi-shell dMRI data from a large cohort of 413 participants to characterize the microstructural properties of the human insular cortex in relation to its major functional subdivisions. There currently are no known landmarks that can distinguish different subdivisions of the insula on the basis of anatomical MRI data. To overcome this problem and provide a better link with its underlying functional organization, we took advantage of recent advances in functional parcellation of the human insular cortex (*Deen et al., 2011*; *Ryali et al., 2015*; *Faillenot et al., 2017*). Resting-state fMRI studies have identified distinct functional subdivisions in the insular cortex based on differential patterns of intrinsic functional connectivity (*Deen et al., 2011*; *Ryali et al., 2015*; *Faillenot et al., 2017*). These subdivisions have been widely used to determine the cognitive and affective role of the insula, and therefore serve as a basis for linking microstructure features of the insula with its functional subdivisions. A widely used parcellation divides the insula into three functional clusters in each hemisphere: dAI, vAI, and PI (*Figure 2*). Our analysis revealed robust evidence for distinct microstructural variations across these functional subdivisions of the insula. Specifically, in both the right and left hemispheres, the lowest RTOP values were in the vAI, followed by the dAI, and the largest in the PI. This pattern is consistent with RTOP profiles in macaque insular cortex (Appendix 2; *Figure 2—figure supplement 1*, *Supplementary file 1*-Supplementary Table 3) and reports of higher density of von Economo neurons in the ventral and dorsal anterior insula, which together form the agranular insular cortex in humans (*Namkung et al., 2017*; *Morel et al., 2013*; *Nimchinsky et al., 1999*). Novel stability analysis revealed that this pattern was reproducible and reliable in samples of 50 or more.

To address the generality of our findings we examined a multimodal HCP surface-based whole-brain parcellation from Glasser and colleagues (*Glasser et al., 2016*), with boundaries similar to but not identical to the functional parcellation from Deen and colleagues (*Deen et al., 2011*). Crucially, all our findings were replicated using this parcellation. Replicability of our findings based on functional subdivisions obtained from two independent studies (*Deen et al., 2011*; *Glasser et al., 2016*), different data and methodological approaches, and stability with respect to sample size in both the primary and replication analyses, emphasizes a close correspondence between functional organization of the insula and its microstructural organization.

## Anterior-posterior and dorsal-ventral gradients in insula microstructure

To further delineate the microstructural organization of the human insula, we conducted a detailed profile analysis to characterize gradients in RTOP along its anterior-posterior and ventral-dorsal axes. The lowest values of RTOP were localized to the vAI, convergent with findings from the analysis of the three functional subdivisions described above. We found strong evidence of an anterior-to-posterior and ventral-to-dorsal gradient in each subdivision of both left and right insula (*Figure 3A–C*). Analysis of microstructure isolines revealed gradient 'fingers' from the vAI extending along a ventral-dorsal axis. This pattern is strikingly consistent with findings from known cytoarchitectonic features in the insula derived from studies of post-mortem brains (*Morel et al., 2013*). The vAI regions where we observed the lowest RTOP is also consistent with histological studies that have reported higher expression of von Economo neurons in this region (*Morel et al., 2013*; *Allman et al., 2010*; *Evrard et al., 2012*; *Figure 4A*). When using the multi-modal parcellation of

Glasser and colleagues (*Glasser et al., 2016*), we also found a gradient spanning from vAI and extending along a ventral-dorsal axis (*Figure 3D–F*). Once more, obtaining a pattern consistent with cytoarchitectonic features of the insula (*Figure 4B*).

The anterior-posterior-ventral-dorsal gradients and convergent findings from the three distinct functional subdivisions provide new quantitative insights into the microstructural organization of the human insular cortex. The precise anatomical boundaries of the agranular, dysgranular, and granular architectonic areas within the human insula are not known, and differences in stereological methods and criteria have led to different segmentation schemes of the insular cortex in primates (*Roberts and Hanaway, 1970*; *Paxinos et al., 2000*). However, there is general consensus that the anterior and most of the mid insula areas are agranular or dysgranular while the posterior most aspects are granular (*Namkung et al., 2017*). The spatial gradient of RTOP in the insular cortex along the dorsocaudal-rostroventral axis provides new metrics for noninvasive in vivo analysis of the general cytoarchitectonic organization in the insular cortex in the human brain and is an advance over previous studies that have thus far been based on invasive histological and electrophysiological data of human and non-human primates (*Morel et al., 2013*; *Brodmann, 1909*; *Mesulam and Mufson, 1985*; *Mesulam and Mufson, 1982b*; *Bonthius et al., 2005*).

## Clusters vs. gradients in relation to insula microstructural and functional organization

Our study focused on a tripartite functional organization of the human insular cortex consisting of dAI, vAI, and PI subdivisions, which is broadly consistent with prior histological reports (*Mesulam and Mufson, 1982b*; *Chikama et al., 1997*; *Brockhaus, 1940*; *Rose, 1928*). This approach assumes a clustered structure; however, depending on the parcellation techniques employed, resting-state fMRI-based studies have reported that the human insula can be divided into anywhere between 2 and 13 distinct subdivisions. Tian and Zalesky have noted that traditional clustering methods applied to insula parcellations based on resting-state fMRI data have several limitations which can be overcome by characterizing the diversity of its functional connectivity along a continuum (*Tian and Zalesky, 2018*; *Tian et al., 2019*). A similar observation has been made in the context of dMRI-derived white matter tracts of the insula (*Cerliani et al., 2012*). Interestingly, this is also the view that emerges from our microstructural characterization of the insula using RTOP which found gradients spanning the anterior-posterior and dorsal-ventral axes, as described in detail above. Taken together, these observations suggest that gradient-based approaches, rather than cluster-based ones, might better reflect the underlying cytoarchitecture of the insula and its connectivity. Further studies using multimodal gradient-based approaches are needed to more precisely probe the association between insula microstructure, connectivity, and function.

## Hemispheric asymmetry in microstructural organization of the human insula

Our analysis of the left and right insula revealed a prominent hemispheric asymmetry in microstructure. RTOP values were significantly lower in the right hemisphere, with the lowest values among all six subdivisions (three in each hemisphere) being localized to the vAI in the right hemisphere. This is consistent with observations in human postmortem data (*Allman et al., 2010*). Interestingly, this asymmetry parallels the differential functional role and engagement of the right AI in cognitive and emotion control tasks such as the Go-Nogo, Stop signal and Emotional Stroop tasks (*Cai et al., 2014*; *Cai et al., 2016*; *Ham et al., 2013*; *Cai et al., 2019*). Consistent with these observations, the right AI has been shown to exert significant causal influences on multiple other brain areas in a wide range of cognitive control tasks (*Cai et al., 2014*; *Cai et al., 2016*; *Ham et al., 2013*; *Chen et al., 2015*) and lesions to this region are known to impair cognitive control (*Ding et al., 2016*). Hemispheric asymmetry in microstructural organization of the insula, and its putative links with differential expression of von Economo neurons, are consistent with reports of a left-right functional asymmetry in the insula (*Craig, 2005*). Specifically, it has been suggested that homeostatic afferent, including hot and cold pain, muscle and visceral pain, sensual touch and sexual arousal all produce strong right-lateralized activation in the right AI (*Craig, 2002*). Heartbeat-related evoked potentials and interoceptive awareness of heartbeat timing, arising from 'sympathetic' homeostatic afferent activity are also associated with AI activity (*Critchley et al., 2004*). We suggest that hemispheric asymmetry

in microstructural organization of the insula may contribute to lateralization of function and in particular the differential role of the right insula in monitoring internal bodily states and subjective awareness across a wide range of cognitive and affective processing tasks (*Craig, 2002*).

## Linked microstructural features in the insula and ACC nodes of the salience

The insular cortex together with the ACC are the two major cortical nodes that anchor the SN (*Menon and Uddin, 2010*; *Seeley et al., 2007*). Noninvasive brain imaging studies using both fMRI and DTI have shown that the insula and ACC are strongly connected functionally and structurally (*Cai et al., 2014*; *Allman et al., 2010*; *Seeley et al., 2007*; *Sridharan et al., 2008*; *van den Heuvel et al., 2009*; *Smith et al., 2009*; *Cauda et al., 2011*), together forming the backbone of the SN. Previous studies have shown that individual functional subdivisions of the insula have preferential connections to different subdivisions of the ACC (*Chang et al., 2013*; *Deen et al., 2011*; *Cauda et al., 2012*; *Taylor et al., 2009*). However, it is not known whether functional subdivisions within the ACC share the similar microstructural properties to the insular subdivisions they connect to. In a significant advance over previous research, we found a close correspondence between microstructural organization of the insula and its interconnected ACC subdivisions. First, using whole-brain functional connectivity analysis we discovered three dorsal ACC subdivisions each with a pattern of preferential functional connectivity with the three insula subdivisions - vAI, dAI and PI. Crucially, as with the three insula subdivisions, we found that these three functionally-defined dorsal ACC subdivisions themselves have distinct microstructural organization. Moreover, the underlying pattern is characterized by a one to one mapping between their respective RTOP values. The ACC subdivision with low RTOP was strongly connected to the insular subdivision with low RTOP, and the ACC subdivision with high RTOP is strongly connected to the insular subdivision with high RTOP. Our findings suggest a strong link between functional circuits linking the insula and ACC and their microstructural organization.

Our findings provide novel insights into the link between the functional organization of the SN and its microstructural features. Although there have been no direct histological investigations of the correspondence between cytoarchitecture features of the human AI and ACC, tract-tracing studies in macaques have shown that the ventral anterior insula receives input from pyramidal neurons in ipsilateral dorsal ACC (*Mesulam and Mufson, 1982b*). Our findings help link microstructural features of the insula with its macroscopic functional connectivity with the ACC for the first time. The correspondence between microstructural and large-scale functional connectivity profiles across the three insula and ACC subdivisions suggests that local and large-scale circuit features may together contribute to the integrity of the SN.

## Insula microstructure predicts cognitive control ability

The human insula has been implicated in a broad range of cognitive functions (*Menon and Uddin, 2010*; *Uddin, 2015*; *Cai et al., 2014*; *Seeley et al., 2012*; *Chen et al., 2016*), but links between its microstructural features and individual differences in cognitive control abilities have not been previously examined. Our analysis revealed an association between the microstructural properties of the insula and individual differences in cognitive control ability. Remarkably, machine learning algorithms and cross-validation leveraging the large HCP sample size also revealed that cognitive control abilities could be predicted in a left-out sample with the strongest predictive weights in the right anterior insula. Our findings provide novel in vivo evidence that the microstructural integrity of the insula is crucial for implementing cognitive control. Gray matter lesions and insults to white matter pathways associated with the insular cortex and pathways linking it to the salience network have also been shown to impair cognitive control ability (*Jilka et al., 2014*; *Clark et al., 2008*; *Gläscher et al., 2012*). In particular, the AI, the key node in the salience network, is one of the most consistently activated brain regions during tasks involving cognitive control (*Cai et al., 2014*; *Swick et al., 2011*). Moreover, the strength of functional and anatomical connectivity between anterior insula and other cognitive control regions is modulated by cognitive demands (*Cai et al., 2016*; *Ham et al., 2013*; *Chen et al., 2015*; *Chen et al., 2016*; *Cai et al., 2017*; *Taghia et al., 2018*; *Wen et al., 2013*). Aberrant connectivity of the anterior insula has been also associated with cognitive deficits in psychiatric disorders (*Cai et al., 2018*; *Uddin et al., 2013*; *Palaniyappan et al., 2013*). Integrating these and

other related findings, a prominent neurocognitive model has proposed that the salience network, especially the anterior insula, plays an important role in dynamically switching between other core brain networks to facilitate access to cognitive resources (*Menon and Uddin, 2010*). A unique feature of the human insular cortex that has been hypothesized to support its role in fast switching is the presence of von Economo neurons, whose large axons could provide a neuronal basis for rapid signal relay between AI and ACC and other brain networks. Our findings provide convergent support for this hypothesis and demonstrate a link between the unique microstructural features of the anterior insula and cognitive control function in humans, and they provide new quantitative metrics for investigating multiple psychiatric and neurological disorders known to impact the insula transdiagnostically (*Goodkind et al., 2015*).

## Conclusion

Our novel quantitative analysis of multi-shell dMRI data provides reliable and replicable noninvasive in vivo measures of gray matter microstructure. We identified several unique microstructural features of the human insula and functional circuits associated with the salience network. Crucially, microstructural properties of the insular cortex are behaviorally relevant as they predicted human cognitive control abilities, in agreement with its crucial role in adaptive human behaviors. Our study provides a novel template for non-invasive investigations of microstructural heterogeneity of the human insula and the salience network, and how its distinct organization may impact human cognition, emotion and interoception. Our findings also open new possibilities for probing psychiatric and neurological disorders that are known to be impacted by insular and cingulate cortex dysfunction, including autism, schizophrenia, depression and fronto-temporal dementia (*Bonthius et al., 2005*; *Allman et al., 2005*; *Brüne et al., 2010*; *Seeley et al., 2006*; *Santos et al., 2011*).

# Materials and methods

**Key resources table**

| Reagent type (species) or resource | Designation | Source or reference | Identifiers | Additional information |
|---|---|---|---|---|
| Other (Human) | Human Connectome Project (HCP) | *Glasser et al., 2013* (DOI: 10.1016/j.neuroimage.2013.04.127) | Dataset: WU-Minn HCP Data - 1200 | |
| Other (Macaque) | NKI PRIMatE Data Exchange database (PRIME) | *Milham et al., 2018* (DOI: 10.1016/j.neuron.2018.08.039) | Dataset: Princeton NA&P Lab | |
| Software, algorithm | DIPY | *Dipy Contributors et al., 2014* (DOI: 10.3389/fninf.2014.00008) | | |

## Human datasets

Data acquisition for the HCP was approved by the Institution Review Board of the Washington University in St. Louis (IRB #201204036), and all open access data were deidentified. Details of the data acquisition, preprocessing and analysis steps are described in the supplementary materials sections B, C, and D see also *Figure 1*.

## Insula ROIs

Our primary analysis focused on an independent tripartite insula parcellation provided by Deen and colleagues (*Deen et al., 2011*; *Figure 2A*). To investigate the generality of our findings, we conducted a replication analysis using the a multimodal HCP-based atlas provided by Glasser and colleagues (*Glasser et al., 2016*). Since the multimodal atlas contains eight insular subdivisions, we reconstructed an equivalent tripartite organization by (i) combining regions AVI+MI+FOP3 into a dAI subdivision, (ii) using AAIC as the corresponding vAI subdivision, and (iii) combining Pol1+Pol2+Ig+FOP2 into a PI subdivision (see Supplementary Materials for details).

## RTOP stability analysis

To evaluate stability of our findings regarding RTOP differences between ROIs, we used subsampling procedures and determined minimum sample sizes that consistently reproduced findings.

## RTOP gradient analysis

To determine whether RTOP values followed an anterior-to-posterior and inferior-to-superior cytoarchitectonic organization of the insular cortex (*Nieuwenhuys, 2012*; *Namkung et al., 2017*; *Morel et al., 2013*), we computed a RTOP gradient field along the insular surface of each participant. To assess that the main gradient directions obtained for each participant were following a common direction, their mean and its confidence interval was computed. Furthermore, we used a Rayleigh test to reject the hypothesis that participants' main gradient directions were uniformity distributed.

## HCP resting-state fMRI data acquisition and processing

Minimal preprocessed resting-state fMRI data (session: rfMRI_REST1_LR) was obtained from the HCP Q1-Q6 Data Release. The resting-state fMRI analysis included 347 subjects from the 433 subjects after head motion screening (total displacement <2 mm, frame-wise displacement <0.2 mm). During scanning, each participant had their eyes fixated on a projected crosshair on the screen. Spatial smoothing with a Gaussian kernel of 6 mm FWHM was first applied to the minimal preprocessed data to improve signal-to-noise ratio as well as anatomy correspondence between individuals. Bandpass temporal filtering (0.008 Hz <f < 0.1 Hz) was then applied.

## fMRI connectivity analysis

Seed-based functional connectivity analyses were conducted to examine whole-brain connectivity patterns of each insular subdivision. First, time series across all the voxels within each insular subdivision were extracted and averaged. The resulting averaged time series was then used as a covariate of interest in a linear regression of the whole-brain analysis. A global time series, computed across all brain voxels, along with six motion parameters were used as additional covariates to remove confounding effects of physiological noise and participant movement. Linear regression was conducted at the individual level. A group map was generated using one-sample $t$-tests (p<0.01, FDR corrected). Last paired $t$ tests were applied at the group level to identify brain regions significantly more correlated with one subdivision than the other (p<0.01, FDR corrected).

FMRI connectivity analysis was used to identify functional subdivisions of the ACC that were differentially connected to each of the insular subdivisions. First, maps of paired comparisons between functional connectivity of different insular subdivisions were thresholded (p<0.01, FDR corrected) and binarized. Second, logical AND operation was applied to identify voxels surviving two paired comparisons for an insular subdivision versus the others (e.g. left PI >left dAI and left PI >left vAI). Last, the resulting mask was overlapped with an ACC mask from Harvard-Oxford Probabilistic Atlas of Human Cortical Area (http://www.cma.mgh.harvard.edu/fsl_atlas.html).

## Relation between insula microstructure and behavior

The relationship between brain and behavioral measures was examined using Canonical Correlation Analysis (CCA) and a cross-validation with prediction approach (*Hotelling, 1992*). Six participants were excluded in the analysis because of missing values in behavioral measures. Brain measures consisted of RTOP values for each subdivision of the insular cortex in each hemisphere (six variables in total). Behavioral measures consisted of in- and out-scanner variables that are highly relevant to cognitive control capacity. In-scanner behavioral task measures included n-back working memory task accuracy and reaction time (RT), relational task accuracy and RT, and gambling task percentage accuracy and RT for larger choice. Out-of-scanner behavioral measures consisted of performance on List sorting, Flanker, Card sorting, Picture sequence tasks from the NIH toolbox, as well as the processing speed. Together, there were 11 behavioral measures. Prediction analysis was performed using leave-one-out cross-validation. Pearson's correlation was used to evaluate the correlation between the predicted brain and behavioral measures.

## Acknowledgements

We thank the editor and two anonymous reviewers for valuable feedback which helped improve the study.

## Additional information

### Funding

| Funder | Grant reference number | Author |
|---|---|---|
| European Commission | NeuroLang - 757672 | Demian Wassermann |
| National Institutes of Health | HD094623 | Vinod Menon |
| National Institutes of Health | HD059205 | Vinod Menon |
| National Institutes of Health | MH084164 | Vinod Menon |
| National Institutes of Health | MH105625 | Weidong Cai |
| INRIA | LargeBrainNets | Demian Wassermann |

The funders had no role in study design, data collection and interpretation, or the decision to submit the work for publication.

### Author contributions

Vinod Menon, Conceptualization, Supervision, Investigation; Guillermo Gallardo, Data curation, Software, Visualization, Methodology; Mark A Pinsk, Resources, Data curation, Methodology; Van-Dang Nguyen, Formal analysis; Jing-Rebecca Li, Formal analysis, Supervision, Investigation; Weidong Cai, Conceptualization, Formal analysis, Validation, Investigation; Demian Wassermann, Conceptualization, Data curation, Software, Formal analysis, Supervision, Validation, Investigation, Visualization, Methodology

### Author ORCIDs

Guillermo Gallardo (iD) https://orcid.org/0000-0002-8271-2516
Weidong Cai (iD) https://orcid.org/0000-0001-9581-7774
Demian Wassermann (iD) https://orcid.org/0000-0001-5194-6056

### Ethics

Human subjects: Data was obtained from the HCP database. Informed consent for this study was not explicitly required. However, subjects signed a written informed consent when the database was constituted. IRB approval was obtained for the database construction with the following details: Mapping the Human Connectome: Structure, Function, and Heritability IRB # 201204036.
Animal experimentation: Animal data was obtained from the INDI-Prime primate data exchange database collection (http://fcon_1000.projects.nitrc.org/indi/indiPRIME.html). All methods and procedures were approved by the Princeton University IACUC.

### Decision letter and Author response

Decision letter https://doi.org/10.7554/eLife.53470.sa1
Author response https://doi.org/10.7554/eLife.53470.sa2

## Additional files

### Supplementary files

• Supplementary file 1. Contains the von Mises-Fisher statistics for the three insula subdivisions dAI, vAI, and PI for each functional parcellation-derived subdivision (*Deen et al., 2011*) in Supplementary Table 1; as well as for each multimodal parcellation-derived subdivision (*Glasser et al., 2016*) in Supplementary Table 2. It also contains the Microstructural properties of insular subdivisions in the macaque brain in Supplementary Table 3. Finally, the sorted weights of the first component in brain-behavior canonical correlation analysis for the analysis using functional parcellations of the insula (*Deen et al., 2011*) is included in Supplementary Table 4; and the result of the same analysis using multimodal parcellations of the insula (*Glasser et al., 2016*) is included in Supplementary Table 5.

• Transparent reporting form

## Data availability

All data used in this study is available in open-source databases. The human data comes from the Human Connectome Project, the primate data is available at the INDI Primate Data Exchange, and the three-dimensional neuronal models are available from the NeuroMorpho website. All custom code is available on GitHub accessible through the Zenodo (DOI: 10.5281/zenodo.3759708). All code was developed based on open-source, publicly available software packages.

The following dataset was generated:

| Author(s) | Year | Dataset title | Dataset URL | Database and Identifier |
|---|---|---|---|---|
| Wassermann D | 2020 | NeuroLang/Microstructural-organization-of-human-insula-is-linked-to-its-macrofunctional-circuitry-and-predicts v0.0 | https://doi.org/10.5281/zenodo.3759708 | Zenodo, 10.5281/zenodo.3759708 |

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

# Appendix 1

## Supplementary methods

### Return to Origin Probability (RTOP) Density of Water Molecules as Measured by Diffusion MRI

In biological tissue, the dMRI signal (*Latour et al., 1994*) and Return-to-Origin probability (RTOP) density are known to be sensitive to volume, surface-to-volume ratio (*Mitra et al., 1995*; *Schwartz et al., 1997*), and mesoscopic organization of cells (*Novikov et al., 2014*). The diffusion MRI signal characterizes the probability that water molecules within a voxel experience, on average, a net displacement after a given diffusion time. This enables the measurement of the probability that, after a diffusion time t, the water particles in a voxel are in the vicinity of their starting position, namely RTOP density (*Mitra et al., 1995*). RTOP has been shown, theoretically and experimentally to be an index closely related, in cellular tissue, to cellular shape and organization (*Mitra et al., 1995*; *Schwartz et al., 1997*). Furthermore, an advantage of RTOP is that it is easy to estimate by integrating dMRI signals in a 3D volume.

The unnormalized RTOP measurement is obtained from the dMRI signal $S$ as:

$$P(t) = \frac{1}{S(0,t)} \int_{R^3} S(q,t) dq$$

In this expression, $P(t)$ is the unnormalized RTOP at a given diffusion time $t$, and $q$ is the wavevector related to the usual diffusion b-value as $q = \sqrt{b}/t$, and the diffusion time is $t = \Delta - \delta/3$ where $\Delta$ and $\delta$ are the gradient pulse separation and length respectively.

Theoretical results enable us to link unnormalized RTOP measurements to tissue microstructure (*Mitra et al., 1995*; *Schwartz et al., 1997*). Specifically, for free diffusion, as in the ventricles, it can be shown that (*Mitra et al., 1995*; *Schwartz et al., 1997*):

$$P(t) = (4\pi D_{vent} t)^{-3/2}$$

where $D_{vent}$ is the ventricular diffusion constant which we estimate by computing the average mean diffusivity within the ventricle.

To render the unnormalized RTOP measurement, $P(t)$, comparable across participants we use its normalized version (*Mitra et al., 1995*).

$$R(t) = (4\pi D_{vent} t)^{3/2} P(t),$$

where $D_{vent}$ is obtained from the ventricles in each participant, by computing the average ventricular mean diffusivity. RTOP, $R(t)$, is now a dimensionless quantity reflecting the relative enhancement of the unnormalized RTOP density, $P(t)$, with respect to free water diffusion (*Mitra et al., 1995*; *Schwartz et al., 1997*).

### HCP diffusion MRI data acquisition and preprocessing

Minimal preprocessed diffusion MRI data (*Glasser et al., 2013*; *Sotiropoulos et al., 2013*) from 433 participants were obtained from the HCP Q1-Q6 Data Release. Twenty participants were excluded in the analysis because of outliers of RTOP values (defined as 1.5 interquartile ranges), leading to 413 subjects in the final sample (22–36 years old, 250 female/172 male). The acquisition protocol was optimized for high spatial resolution and signal-to-noise ratio. Two datasets with LR and RL polarities were acquired in each participant: TE/TR 89 ms/5.5 s; 1.25 mm isotropic voxels; b = 0 (6 images), and 271 directions equally distributed across 3 b-shells of 1, 2, and 3 ms $\mu m^{-2}$; diffusion times $\Delta$=43.1 ms and $\delta$ = 10.6 ms. We refer the interested reader to *Sotiropoulos et al. (2013)* for further detail of the acquisition and preprocessing pipeline.

## Computing RTOP from diffusion MRI data

The dMRI signal characterizes the probability that water molecules within a voxel experience a net displacement after a given diffusion time. This enables the measurement of the probability that, after a diffusion time t, water in a voxel are in the vicinity of their starting position, namely the RTOP density (*Mitra et al., 1995*). We computed RTOP at the voxel level on the preprocessed diffusion MRI images provided by the HCP. At each voxel, we computed RTOP by evaluating a regularized representation of the signal based on the Mean Average Propagator formalism with Laplacian regularization (*Fick et al., 2016*) (MAPL) included in the dipy open-source software package (*Dipy Contributors et al., 2014*) (http://nipy.org/dipy). The regularization parameter was selected through generalized cross validation and RTOP was computed analytically from the fitted MAPL parameters. Finally, to render the Return-To-Origin probability comparable across subjects, we normalized it by the average ventricular Return-To-Origin probability of each subject's cortico-spinal fluid in the ventricles. The normalized RTOP quantifies the enhancement ratio over free diffusion (*Mitra et al., 1995*; *Schwartz et al., 1997*) specific to each participant (Appendix 1). Furthermore, the normalized RTOP is sensitive to the volume and distribution of cell types across the insula, thereby allowing us to capture probe insular cortex cytoarchitectonic organization (Appendix 2.

## Projecting RTOP values to the template cortical surface

To obtain the values of RTOP on a template cortical surface we started by sampling the voxel-level RTOP on to the participant-specific 32k cortical surfaces. We used the provided mid-thickness surfaces. This was done with the objective of projecting onto the surface the interpolated voxel RTOP value at the mid-point between the pial surface and the grey-white matter interface. Finally, to bring the surface-projected RTOP values to the common HCP template space we used the correspondence between the subject-specific surfaces and the template-registered MSMAll surfaces (*Glasser et al., 2013*).

## Human insula and anterior cingulate cortex ROIs

We used functional parcellation of insular cortex from an independent study (*Deen et al., 2011*) to study RTOP variability between insular subregions. This parcellation of the insula includes ventral anterior insula (vAI), dorsal anterior insula (dAI) and posterior insula (PI).

To replicate our findings, we used the multimodal whole-brain atlas derived from a large number of HCP participants (*Glasser et al., 2016*). We constructed an equivalent tripartite organization we defined as a homologous set of dAI, vAI and PI subdivisions by (i) combining regions AVI+MI+FOP3 into a dAI subdivision, (ii) using AAIC as the corresponding vAI subdivision, and (iii) combining Pol1+Pol2+Ig+FOP2 into a PI subdivision. We projected the resulting regions to each participants' cortical surface by leveraging the fact that surfaces are coregistered across subjects in the HCP dataset (*Glasser et al., 2013*).

Anterior cingulate cortex (ACC) ROIs were demarcated using the Harvard-Oxford atlas (*Desikan et al., 2006*). We also examined whether ACC subdivisions defined by their distinct functional connectivity with the insular subdivisions, matched distinct ACC areas p24, a24pr and p24pr in the HCP multimodal atlas (*Glasser et al., 2016*).

## Macaque MRI dataset

We used data from two male rhesus macaques (Macaca mulatta). The Princeton University Animal Care and Use Committee approved all procedures, which conformed to the National Institutes of Health guidelines for the humane care and use of laboratory animals.

## Macaque MRI data acquisition and processing

We scanned two male rhesus macaques (Macaca mulatta, ages = 4 years, body weights = 5.6/6.8 kg). For all scan sessions, animals were first sedated with ketamine (10 mg/kg IM) and maintained with isoflurane gas (2.5–3.0%) using an MR-compatible anesthesia workstation (Integra SP II, DRE Inc, Louisville KY). Data were collected to resemble the HCP dMRI protocol as close as possible. Specifically, we acquired the whole brain on a Siemens 3T MAGNETOM Prisma (80 mT/m @ 200 T/m/s gradient strength) using a surface coil (11 cm Loop Coil; Siemens AG, Erlangen Germany) secured above the head. Nine T1-weighted volumes were collected for averaging to obtain a high-quality structural volume and diffusion images were acquired using a double spin-echo EPI readout pulse sequence. Two datasets per macaque were acquired with opposite LR, RL polarity: resolution 1 mm isotropic; TE/TR 83 ms/8 s; 50 slices; FOV/Matrix 96 mm/96; 18 images at b = 0; and 271 directions equally distributed at b = 0.85, 1.65, and 2.5 ms $\mu m^{-2}$ where the gradient directions were distributed optimally across the three spherical shells; diffusion times $\Delta$=36.8 ms and $\delta$ = 14.1 ms. Distortion correction was applied using FSL's Eddy tool. All macaque MRI data are publicly available on the NKI PRIMatE Data Exchange database (*Milham et al., 2018*). The T1-weighted images were registered to the D99 atlas through the NMT—NIMH Macaque Template allowing to subdivide the insula into granular, dysgranular, and agranular areas along with smaller subdivisions of each parcel. RTOP was computed using similar procedures as the human dMRI data.

## Appendix 2

## Supplementary results

### RTOP captures known microstructural features of the insular cortex in macaques

Previous histological studies have demonstrated that the macaque insula can be clearly demarcated into agranular, dysgranular and granular subdivisions based on their unique cytoarchitectural properties (*Evrard et al., 2012*; *Evrard et al., 2014*; *Figure 2—figure supplement 1*). To investigate whether RTOP measures are sensitive to known cytoarchitectural features of the primate insula, we acquired dMRI data from two macaque monkeys, X77 and X181, using protocols similar to the HCP, and examined RTOP values in the known cytoarchitectonic subdivisions of the primate insula. We found that in both macaques, the lateral agranular insula (Ial) was the region with the lowest RTOP values. Specifically, in animal X77, the RTOP values inside the right Ial were significantly lower than other insular regions (*Figure 2—figure supplement 1B*, *Supplementary file 1*-Supplementary Table 3). In X77's left hemisphere, the Ial had significantly lower RTOP than all other regions (two-sided t-test) except Agranular Insula (Ia) and the posterior-medial agranular insula (Iapm) (*Figure 2—figure supplement 1B*, *Supplementary file 1*-Supplementary Table 3). In X181, the left Ial showed significantly lower RTOP than other insular regions (*Figure 2—figure supplement 1B*, *Supplementary file 1*-Supplementary Table 3). Finally, the right Ial had lower RTOP than other regions, except for the Ia (*Figure 2—figure supplement 1B*, *Supplementary file 1*-Supplementary Table 3). Crucially, the granular insular cortex (Ig) contained the highest RTOP values in both monkeys. Ig also has significantly higher RTOP values than other insular regions, with the exception of the left intermediate agranular insula (Iai) in X77 (*Supplementary file 1*-Supplementary Table 3). These results demonstrate that the agranular insula shows the lowest RTOP, demonstrating converging findings with known cytoarchitectonic studies of the insula in macaques.

### RTOP and insula microstructure

Here we present simplified theoretical back-of-the-envelope calculations relating RTOP to insula microstructure. At long diffusion times RTOP can be simplified: if the compartment is restricted along $d$ dimensions, the unnormalized RTOP converges asymptotically to

$$P(t)l^{-d}(D_{intra}t)^{(d-3)/2}$$

where $D_{intra}$ is the water diffusion constant within the compartment and $l$ is the average characteristic length scale of the compartment. For a sphere-type restriction, which models a simplified cellular soma, there are no unrestricted dimensions, hence d=3 and $l^3 \propto V$ is the soma's internal volume, hence, when $\sqrt{6D_{intra}t} \gg l, P(t)l^{-3} \propto V^{-1}$.

The specific properties of insular tissue are consistent with these approximations: first, the long diffusion time bound can be estimated by approximating the intra-cellular water diffusivity as $D_{intra} \sim 3\mu m^2 ms^{-1}$ and taking into account the HCP diffusion times of t=40ms, then $\sqrt{6D_{intra}t} \sim 27\mu m$; second, the average external diameter for the largest neurons in the insula is of l24μm (*Evrard et al., 2012*), leading to a smaller internal diameter. Hence, the expression of the soma component of our normalized RTOP measure in the insula can be approximated as

$$R(t) \propto V^{-1}(D_{vent}t)^{\frac{3}{2}}$$

To determine the extent to which the $R(t)$ is dominated by the soma, as opposed to axonal and dendritic processes, we obtained three-dimensional reconstructions of *Allman et al. (2010)* insular neurons from NeuroMorpho.org. We then computed, on a per neuron basis, the ratio of volume between soma and axonal/dendritic processes obtaining an average ratio of 3.

Using the biophysical model by *Veraart et al. (2019)*, with intra-cellular water diffusivity of $D_{intra} \sim 3\mu m^2 ms^{-1}$ at a maximum b-value of 3 ms $\mu m^{-2}$, we can estimate that less that 10% of $R(t)$'s intracellular signal is modulated by axonal and dendritic processes.

Assuming that extra-cellular compartment volume remains constant, these analyses suggest that RTOP is sensitive to volume and distribution of cell types across the insula and that RTOP reflects key aspects of variation in insular cortex cytoarchitectonic organization. Detailed computer simulations that take into account known layer specific distributions of neurons and glia are needed to further validate these theoretical calculations.

