## [Decision Letter]

**Acceptance summary:**

The paper demonstrates the efficacy of a new non-invasive method that is sensitive to histological features such as cell size. This is an important advance that opens up studies, for example, in other brain regions and clinical populations. It provides a new level of detail in our understanding of the microstructure of the human insula. The study is further unusual in providing comparison between human and nonhuman primate. This is important as much of our knowledge about the cytoarchitecture and connectivity in general, and in insula in particular, is derived from extrapolations from other primate species.

**Decision letter after peer review:**

Thank you for submitting your article "Microstructural organization of human insula is linked to its macrofunctional circuitry and predicts cognitive control" for consideration by *eLife*. Your article has been reviewed by two peer reviewers, and the evaluation has been overseen by Timothy Behrens as the Senior and Reviewing Editor. The following individuals involved in review of your submission have agreed to reveal their identity: Matthew F Glasser (Reviewer #1); Leonardo Cerliani (Reviewer #2).

The reviewers have discussed the reviews with one another and the Reviewing Editor has drafted this decision to help you prepare a revised submission.

Summary:

The authors provide an interesting analysis of the relationships between microstructure, functional connectivity and behavioural specialization for different regions of the human insular cortex. The most original finding relates to the distribution of a DWI-derived microstructural index denoted RTOP (Return to Origin Probability), which is sensitive to the volume and distribution of cell types in different cortical regions, and to the relationship of microstructural feature and functional specialization in insular and functionally connected cingulate regions, together representing the core of the Salience network. The major contributions of the work are a demonstration of sensitivity to histological features such as cell size in vivo with an innovative MR technique, a demonstration of how these features change in the human insular cortex, and a demonstration of the relationship between these features across distant, but connected, regions. The authors should also be commended for replicating the RTOP analysis in specimens of nonhuman primates, as most of our knowledge about insular cytoarchitecture and connectivity still relies on extrapolation from other primate species.

Essential revisions:

In the below, I have replicated comments from the original reviews, but I have previewed them with key elements of our discussion where we tried to clarify which comments should be addressed directly and which could be discussed.

Insular Parcellation:

Several concerns were raised about the insular cortex parcellation that you use. These concerns were about (i) the quality, and (2) the hemispheric asymmetry of the particular parcellation that you use, which could affect the results and (3) the group- rather than individual- nature of the parcellation, in the face of large reported individual differences. Reviewer comments are copied below. In the discussion, the reviewers agreed that it would be sufficient to reproduce key results using the HCP parcellation (described in first comment below), and to discuss the potential impact of individual variability in the discussion. It is hoped that the HCP parcellation (which does not contain the clear asymmetry present in the Deen parcellation) will improve some puzzles in the current results (eg pronounced asymmetry in Figure 2C). We hope that this will be straightforward as it simply requires running your existing pipeline on a new set of insular masks that are freely available.

Comment 1:

"The insular parcellation used by the authors appears of poor quality. For one thing, it is a volume-based parcellation generated from a small number of subjects (n=30) using a clustering algorithm (whose winner take all hard parcellation will be sensitive to noise when near the boundary) only on resting state data. The HCP has produced a much higher quality multi-modal parcellation of the whole cerebral cortex (Glasser et al., 2016 see Supplementary Neuroanatomical Results Figure 14), including the insula, from 210 HCP subjects and precisely replicated it in an additional 210 HCP subjects. The HCP recommends the use of this parcellation together with HCP data. Notably, this parcellation does not show the large hemispheric asymmetry found in the insular parcellation used by the authors in either the final parcellation or the multi-modal data it was defined from, despite the fact that such asymmetries could be found (and in fact were found) in some parts of the cortex in the HCP parcellation. Thus, any of the asymmetries found by the authors using the parcellation may result from idiosyncrasies of that parcellation (e.g. the failure to separate right dAI vs vAI, which I didn't see mentioned anywhere). That said, the three regions described by the authors might be reconstructed from the HCP's parcellation by grouping AAIC (vAI), AVI+MI(dAI), and Pol1+Pol2+Ig(PI) if desired. Depending on how the authors want to define the insular cortex, they might also include FOP3 and FOP2 in dAI and π respectively. Similarly, if the analysis were done with an HCP-Style approach (Glasser et al., 2016) the authors might find that their anterior cingulate regions match areas p24, a24pr, and p24pr (Glasser et al., 2016, see Supplementary Neuroanatomical Results Figure 19). Of note, the agranular to granular transition in the anterior cingulate cortex is much more pronounced in the inferior to superior direction than the anterior to posterior direction (Glasser et al., 2016 Supplementary Neuroanatomical Results section 19, Glasser and Van Essen, 2011, Journal of Neuroscience section on Cingulate Cortex)."

Comment 2: (Which should be addressed in discussion and not in new analyses.)

Interindividual variability and delimitation of insular regions.

The Authors state that "no consensus has yet emerged about their precise boundaries because of small sample sizes, limited insular divisions examined and high degree of variability across individuals". The first rationale of the statement – the small sample size of the histological samples – is not necessarily pertinent to accurately localize cytoarchitectonic transitions. For instance the transition between primary sensory and motor cortex is always very stablly localized, while other transitions – e.g. between BA44 and BA45 – are also clear at a microscopic examination, but present high intersubjective variability in terms of localization (cfr Amunts K et al., 1999 J Comp Neurology). In fact, the latter – rather than small sample sizes – could represent the main reason why also in the insula it is difficult to distinguish different architectural territories. Given this intersubjective variability – as mentioned by the Authors – it is peculiar that the present investigation relies on an insular parcellation which does not estimate such interindividual differences, as it was extracted from the mean connectivity matrix (cfr. Deen et al., 2010 p. 2).

In light of these considerations, I believe that the present analysis would have been much more compelling and data-driven if instead of using a predefined parcellation, the Authors would have carried out a tripartite parcellation of the insula in every individual using the rs-fMRI data at hand, even just with a simple k-means. Since the rs-fMRI data has already been preprocessed for the functional connectivity analysis, such procedure would have been quite easy to implement. I understand that this would probably be an excessive burden to carry out since the insular parcellation represents the very first step of the analysis, but I would like to ask the Authors to acknowledge the limitations the current choice.

Reference to prior work in insular parcellation:

There were several comments about your scholarship with reference to prior work in insular parcellation (some of this work is referenced in the comments above and others in the comments I have copied here)

"Further, there are multiple statements in the paper about lack of prior non-invasive mapping of the insula in the paper that must be toned down in the face of extensive prior literature."

"The Authors present the results by Deen, 2010 and Chang, 2012 as the reference for insular parcellation. They should mention that also other parcellations have been proposed based on similar data, e.g. Jakab et al., 2012; Cauda et al., 2010; Kelly et al., 2012. Importantly, most of these work do not report that a tripartite parcellation represents the best solution for the insula.

As a side note, reference work for human – vs other primates – insular parcellation are not cited. See e.g. the work of Rose and Brockhaus in the review by Nieuwenhuys."

"Correct the neuroanatomical statements in the paper, since they contain several claims which diverge from the reference literature.

"The dysgranular cortex has an intermediate profile that has been mainly observed in the dorsal anterior aspects of the insula". This is not entirely correct, and in this form it is misleading. The dysgranular architecture characterizes as well the central and ventral territory of the insula. Indeed, most of the insular cortex can be characterized as dysgranular. The results in the cited reference (Kurth et al., 2010) show that the ventral middle part of the insula has a dysgranular architecture, while they do not provide results regarding the anterior dorsal territory.

"The anterior insula is more strongly connected to brain areas important for cognitive control, most notably the dorsal anterior cingulate cortex (ACC) while the posterior insula has stronger links with subcortical and limbic regions important for emotion, including the amygdala and ventral striatum". This statement diverges from the reference literature: the amygdaloid complex and the ventral striatum are mostly connected with the ventral anterior insula (cfr. Amaral and Price, 1984 J Comp Neurol and Chikama et al., 1997, respectively), while these connections decrease along the antero-posterior axis."

Data processing:

The reviewers raised concerns that the data processing is not state of the art. This was a surprise as the HCP data that the authors have used is freely available in processed form with state of the art techniques. The relevant reviewer comment is copied below. In discussion, the reviewers agreed that we should strongly encourage the authors to redo analyses as described in the comment. We think this represents best practice. However, we realise that this means redoing the analyses from scratch and, although, we believe it would improve results and represent best practice, we do not think it would fundamentally change the key message of the paper. Hence, in line with *eLife* review policy, this is therefore a recommendation not a requirement for publication.

Reviewer comment:

"It has been replicated repeatedly that cross-subject cortical alignment and ability to localize signals to cortical areas is maximized by using surface-based intersubject registration instead of volume-based registration and, in particular, by avoiding 3D unconstrained volume-based smoothing (Reviewed Glasser et al., 2016 Nature Neuroscience; explicitly redemonstrated Coalson et al., 2018 PNAS). As such, the HCP has provided precisely aligned surface-based CIFTI resting state data (aligned with the MSMAll algorithm). Thus, it was puzzling to find that the authors choose to start from the less well aligned volume-based data and then smoothed the data with "a Gaussian kernel of 6mm FWHM was first applied to the minimal preprocessed data to improve signal-to noise ratio as well as anatomy correspondence between individuals." Importantly, the second statement is a common brain imaging misconception that has been decisively disproved by Coalson et al., 2018 PNAS. Further, the fMRI denoising approach chosen by the authors does not match HCP recommendations-rather they should use the data denoised by spatial ICA and the FIX classifier and do not need to remove movement regressors again. Temporal filtering is also discouraged by the HCP as it affects both signal and noise equally. Further, for the purposes of this study, the authors can easily make use of the publicly available group average MIGP resting state data, where the alignment, structured noise denoising, and significant unstructured noise denoising have already been taken care of-if they do not wish to worry about these methodological details. "

Reporting of methodological detail.

There were several concerns related to the reporting of methodological detail that need to be addressed.

The description of key imaging methods is inadequate. It is useful to restate the key aspects of the HCP diffusion acquisition and preprocessing for readers so they need not refer back to another paper. Additionally, the macaque data acquisition description, apparently new data acquired for the paper, is inadequate. For example, the resolution, TR/TE, b-shells, and preprocessing details are left unspecified. Of note the HCP data were acquired with a single spin echo sequence to enable minimizing the TE and maximizing the SNR. The preprocessing of the macaque diffusion data needs to be described. Further, no details are given as to how the surface-based maps of RTOP were created and any additional processing (e.g. spatial smoothing, cross-subject alignment) that was done on the HCP preprocessed diffusion data or derivatives of this. Depending on the preferred approach of the authors (ideally the one that worked the best) the RTOP data should have been computed in the individual subject's physical volume space and then mapped from between the individual's MSMAll aligned white matter and pial surface meshes onto the 32k MSMAll surface standard space without any spatial smoothing or after mapping the preprocessed diffusion data from between the individual's MSMAll aligned white matter and pial surface meshes onto the 32k MSMAll surface standard space without any spatial smoothing and then modeling RTOP. This data could then simply be averaged across subjects. With 449 subjects, smoothing is unlikely to be necessary for SNR purposes, but modest surface-based smoothing reduces precision much less than volume-based smoothing (Coalson et al., 2018 PNAS).

More information about the CCA analysis.

The association between RTOP and cognitive control measures represents an important part of the results. While the correlation between the left and right canonical vectors is significant and features a moderate effect size, I have some issues with the interpretation of the weights in the first component of the CCA, especially because I know little about this technique.

1) The right dorsal anterior insula has the highest weight, as expected and detailed in the Discussion. However it is a negative weight, while other weights, and notably the rvAI, are positive. What is the meaning of the sign and of the sign difference in the weights, in the interpretation of the microstructure-behavioural association?

2) The weights of the first behavioural canonical variate appear flat, and most of them are zero. What are the implications of this for the interpretations?

3) It would be useful to have a plot of the cumulative variance explained e.g. by the first N pairs of canonical variate, to assess whether a different combination of regional microstructur/behaviour adds a substatial amount of variance

4) Does the significance of the correlation between the first pair of canonical variates survive if permutation testing is used instead of parametric testing?

Introduce the RTOP measure more clearly to a non-specialised audience:

"Explain neuroanatomical correlates of RTOP, which is also useful to interpret the behavioural correlation with microstructure. Differences in insular microstructure are quantified using the RTOP DWI derivative. However I couldn't find an explanation of the microstructural properties that the RTOP targets in the Introduction or in the Results, with only one small mention at the end of the Supplementary Materials. Think e.g. about fractional anisotropy (FA): a neuroanatomist would like to know not only that there are localized differences in FA between samples of participants, but also how FA relates to the microstructure of axonal integrity and myelin sheet, in order to interpret those differences. Please include a brief description of the microstructural properties that RTOP allows to highlight, and the rationale that links these properties to the RTOP calculation. Preferably this should be present in the Introduction."

---

## [Author Response]

[…] Comment 1:"The insular parcellation used by the authors appears of poor quality. For one thing, it is a volume-based parcellation generated from a small number of subjects (n=30) using a clustering algorithm (whose winner take all hard parcellation will be sensitive to noise when near the boundary) only on resting state data. The HCP has produced a much higher quality multi-modal parcellation of the whole cerebral cortex (Glasser et al., 2016 see Supplementary Neuroanatomical Results Figure 14), including the insula, from 210 HCP subjects and precisely replicated it in an additional 210 HCP subjects. The HCP recommends the use of this parcellation together with HCP data. Notably, this parcellation does not show the large hemispheric asymmetry found in the insular parcellation used by the authors in either the final parcellation or the multi-modal data it was defined from, despite the fact that such asymmetries could be found (and in fact were found) in some parts of the cortex in the HCP parcellation. Thus, any of the asymmetries found by the authors using the parcellation may result from idiosyncrasies of that parcellation (e.g. the failure to separate right dAI vs vAI, which I didn't see mentioned anywhere). That said, the three regions described by the authors might be reconstructed from the HCP's parcellation by grouping AAIC (vAI), AVI+MI(dAI), and Pol1+Pol2+Ig(PI) if desired. Depending on how the authors want to define the insular cortex, they might also include FOP3 and FOP2 in dAI and π respectively. Similarly, if the analysis were done with an HCP-Style approach (Glasser et al., 2016) the authors might find that their anterior cingulate regions match areas p24, a24pr, and p24pr (Glasser et al., 2016, see Supplementary Neuroanatomical Results Figure 19). Of note, the agranular to granular transition in the anterior cingulate cortex is much more pronounced in the inferior to superior direction than the anterior to posterior direction (Glasser et al., 2016 Supplementary Neuroanatomical Results section 19, Glasser and Van Essen, 2011, Journal of Neuroscience section on Cingulate Cortex)."

Thank you for raising these important points. Following these suggestions, we have now conducted a replication analysis using the HCP multimodal parcellation^35^. These results are now presented in the manuscript. Crucially, all major findings were replicated with the HCP multimodal parcellation. Our approach avoids potential circularity of using multimodal HCP parcellations to probe insular organization, while also demonstrating replicability.

To examine reproducibility of the insula microstructure across functionally defined subdivisions, we conducted the same analysis using the independent insular parcellation from Glasser et al^35^ (Figure 2D). We found a significant interaction between hemisphere and subdivision (ANOVA, F(2,824) = 207.3, *p* < 2E-16) and significant main effects of subdivision (ANOVA, F(2,824) = 11265, *p* < 2E-16) and hemisphere (ANOVA, F(1,412) = 10.9, *p* < 0.001) (Figure 2E). Post-hoc *t*-tests further revealed a gradient of RTOP values vAI < dAI < π in both right and left hemisphere (all *ps* < 3.2E-06, with vAI and π having significantly smaller values in right hemisphere and vAI having significantly smaller values in the left hemisphere (paired t-test, all *ps* < 3.2E-06) (Figure 2E). Stability analysis revealed that a sample size of N=25 was sufficient to achieve a stable differentiation (*p* <0.01) between π and vAI and between π and dAI in both hemispheres and between dAI and vAI in the left hemisphere (Figure 2F).

To examine reproducibility of microstructural features in ACC nodes of the salience network, we extended the analysis using ACC subdivisions determined by their functional connectivity with the insula parcellation of Glasser et al^35^. We found a significant interaction between ACC subdivision and hemisphere (ANOVA, F(2,824) = 245.7, *p* = 2E-16), and significant main effects of subdivision (ANOVA, F(2,824) = 343, *p* = 2E-16) and hemisphere (ANOVA, F(1,412) = 248.9, *p* = 2E-16) (Figure 7E). Post-hoc paired *t*-tests revealed significant differences in RTOP: ACC-vAI < ACC-dAI < ACC-PI in both hemispheres (paired t-tests, all *ps* < 6E-11) (Figure 7E). Stability analysis demonstrated that the differences between π and vAI and between dAI and vAI in both hemispheres and between π and dAI in the right hemisphere were highly reliable (paired t-tests, *p* < 0.01) for samples sizes > N=50 (Figure 7F). The three ACC subdivisions overlapped with distinct areas demarcated by the HCP multimodal atlas^35^: the ACC-vAI overlapped with ACC area p24, ACC-dAI overlapped with cingulate area ap24pr, and ACC-PI overlapped with ACC area p24pr, thus demonstrating distinct patterns of connectivity linking insula and ACC subdivisions.

To examine reproducibility of insula microstructure in relation to cognitive control ability, we conducted the same analysis using the insular parcellation from Glasser et al^35^. We found a marginally significant relation between insula RTOP measures and individual cognitive control abilities (Canonical correlation, Pillai’s trace = 0.20, *p* = 0.08). The canonical weights of the 1^st^ latent variable in microstructural measures were significantly correlated with the canonical weights of the 1^st^ latent variable in behavioral measures (*Pearson*’s correlation, *r* = 0.31, *p* < 0.001, *Cohen’s d* = 0.65, Figure 8C; Supplementary file 1—supplementary table 5). A leave-one-out cross-validation procedure further revealed that, based on microstructural properties of the insula, our CCA model could predict cognitive control ability on unseen data (*Pearson*’s correlation, *r* = 0.17, *p* < 0.001, *Cohen’s d* = 0.35, Figure 8D ), with the strongest predictive weights in the right dAI.

In summary, these results demonstrate replication of our previous findings using the HCP parcellation.

Comment 2: (Which should be addressed in discussion and not in new analyses.)Interindividual variability and delimitation of insular regions.The Authors state that "no consensus has yet emerged about their precise boundaries because of small sample sizes, limited insular divisions examined and high degree of variability across individuals". The first rationale of the statement – the small sample size of the histological samples – is not necessarily pertinent to accurately localize cytoarchitectonic transitions. For instance the transition between primary sensory and motor cortex is always very stablly localized, while other transitions – e.g. between BA44 and BA45 – are also clear at a microscopic examination, but present high intersubjective variability in terms of localization (cfr Amunts K et al., 1999 J Comp Neurology). In fact, the latter – rather than small sample sizes – could represent the main reason why also in the insula it is difficult to distinguish different architectural territories. Given this intersubjective variability – as mentioned by the Authors – it is peculiar that the present investigation relies on an insular parcellation which does not estimate such interindividual differences, as it was extracted from the mean connectivity matrix (cfr. Deen et al., 2010 p. 2).In light of these considerations, I believe that the present analysis would have been much more compelling and data-driven if instead of using a predefined parcellation, the Authors would have carried out a tripartite parcellation of the insula in every individual using the rs-fMRI data at hand, even just with a simple k-means. Since the rs-fMRI data has already been preprocessed for the functional connectivity analysis, such procedure would have been quite easy to implement. I understand that this would probably be an excessive burden to carry out since the insular parcellation represents the very first step of the analysis, but I would like to ask the Authors to acknowledge the limitations the current choice.

We thank the reviewers for this suggestion. We appreciate that there are several reasons why no consensus has yet emerged about insular subdivisions, and note that the goal of our study was not to resolve potential issues of variability in insular boundaries. Rather, our goal was to use generally accepted consensus parcellations to probe insular microstructure. We have now therefore revised the statement to read: “In the ensuing years, several histological studies have focused on demarcating the microstructural properties of insular subdivisions but no consensus has yet emerged about their precise boundaries”. We agree with the sentiment of the reviewer that individual subject parcellations may have the potential to provide additional insights into insular functional organization. It may be noted, however, that while the use of individual parcellations has its strengths, there are also limitations as different participants might have different numbers of optimal (k-means-derived) clusters, making it difficult to match parcellations across individuals. Please also see reply below regarding gradients vs. clusters.

Importantly, to address the generality of our findings we examined an independent surface-based parcellation^35^, albeit with boundaries similar but not identical to the functional parcellation of Deen and colleagues^29^. Crucially, all our findings were replicated using the multimodal parcellation. The replicability our findings based on functional subdivisions obtained from two independent studies^29,35^, based on different data and methodological approaches, and stability with respect to sample size in both the primary and replication analyses thus points to a close correspondence between functional organization of the insula and its microstructural organization.

Reference to prior work in insular parcellation:There were several comments about your scholarship with reference to prior work in insular parcellation (some of this work is referenced in the comments above and others in the comments I have copied here)"Further, there are multiple statements in the paper about lack of prior non-invasive mapping of the insula in the paper that must be toned down in the face of extensive prior literature.""The Authors present the results by Deen et al., 2010 and Chang et al., 2012 as the reference for insular parcellation. They should mention that also other parcellations have been proposed based on similar data, e.g. Jakab et al., 2012; Cauda et al., 2010; Kelly et al., 2012. Importantly, most of these work do not report that a tripartite parcellation represents the best solution for the insula.As a side note, reference work for human – vs other primates – insular parcellation are not cited. See e.g. the work of Rose and Brockhaus in the review by Nieuwenhuys."

We apologize for the confusion here. Please note that our statements about lack of prior non-invasive mapping of the insula did not refer to resting-state fMRI studies of which there are many. Rather, our statements referred to lack of non-invasive mapping studies on microstructural properties of the insula. We have gone over all such statements and ensured that this is indeed the case.

We apologize for the oversight and have now cited the important work of Jakab et al., Cauda et al. and Kelly et al. in the Introduction. Depending on the parcellation techniques employed, previous studies have reported that the human insula can be divided into anywhere between 2 and 13 distinct subdivisions. Most studies, however, point to distinct dorsal anterior insula, ventral anterior insula, and posterior insula segments, consistent with histological reports^26^ Ultimately, as noted by Tjan, Zalesky and colleagues^64,65^, clustering methods are not ideal for studying the insula, and may need to be replaced with gradient based analyses. Please see reply below for further details.

We have also now cited the important original work of Rose and Brockhaus, and thank the reviewer for pointing them out to us.

"Correct the neuroanatomical statements in the paper, since they contain several claims which diverge from the reference literature."The dysgranular cortex has an intermediate profile that has been mainly observed in the dorsal anterior aspects of the insula". This is not entirely correct, and in this form it is misleading. The dysgranular architecture characterizes as well the central and ventral territory of the insula. Indeed, most of the insular cortex can be characterized as dysgranular. The results in the cited reference (Kurth et al., 2010) show that the ventral middle part of the insula has a dysgranular architecture, while they do not provide results regarding the anterior dorsal territory.

We have corrected this statement to read “The dysgranular cortex has an intermediate profile that has been observed in the anterior aspects of the insula”. We thank the reviewer for this feedback.

"The anterior insula is more strongly connected to brain areas important for cognitive control, most notably the dorsal anterior cingulate cortex (ACC) while the posterior insula has stronger links with subcortical and limbic regions important for emotion, including the amygdala and ventral striatum". This statement diverges from the reference literature: the amygdaloid complex and the ventral striatum are mostly connected with the ventral anterior insula (cfr. Amaral and Price, 1984 J Comp Neurol and Chikama et al., 1997, respectively), while these connections decrease along the antero-posterior axis."

We have now clarified the distinction between the dorsal and ventral anterior insula and now state: “This functional organization is supported by a distinct pattern of long-range connections: the dorsal anterior insula is more strongly connected to brain areas important for cognitive control, most notably the dorsal anterior cingulate cortex (ACC) while the ventral anterior insula and posterior insula have stronger links with subcortical and limbic regions important for emotion, reward and homeostatic regulation, including the amygdala, ventral striatum and hypothalamus^2,4,25,26^. Consistent with these reports, meta-analysis of task-related coactivation patterns in human neuroimaging studies point to distinct functional networks associated with insular subdivisions^27^”.

Data processing:The reviewers raised concerns that the data processing is not state of the art. This was a surprise as the HCP data that the authors have used is freely available in processed form with state of the art techniques. The relevant reviewer comment is copied below. In discussion, the reviewers agreed that we should strongly encourage the authors to redo analyses as described in the comment. We think this represents best practice. However, we realise that this means redoing the analyses from scratch and, although, we believe it would improve results and represent best practice, we do not think it would fundamentally change the key message of the paper. Hence, in line with eLife review policy, this is therefore a recommendation not a requirement for publication.Reviewer comment:"It has been replicated repeatedly that cross-subject cortical alignment and ability to localize signals to cortical areas is maximized by using surface-based intersubject registration instead of volume-based registration and, in particular, by avoiding 3D unconstrained volume-based smoothing (Reviewed Glasser et al., 2016 Nature Neuroscience; explicitly redemonstrated Coalson et al., 2018 PNAS). As such, the HCP has provided precisely aligned surface-based CIFTI resting state data (aligned with the MSMAll algorithm). Thus, it was puzzling to find that the authors choose to start from the less well aligned volume-based data and then smoothed the data with "a Gaussian kernel of 6mm FWHM was first applied to the minimal preprocessed data to improve signal-to noise ratio as well as anatomy correspondence between individuals." Importantly, the second statement is a common brain imaging misconception that has been decisively disproved by Coalson et al., 2018 PNAS. Further, the fMRI denoising approach chosen by the authors does not match HCP recommendations-rather they should use the data denoised by spatial ICA and the FIX classifier and do not need to remove movement regressors again. Temporal filtering is also discouraged by the HCP as it affects both signal and noise equally. Further, for the purposes of this study, the authors can easily make use of the publicly available group average MIGP resting state data, where the alignment, structured noise denoising, and significant unstructured noise denoising have already been taken care of-if they do not wish to worry about these methodological details."

We appreciate the recommendations of the reviewers and thank the editors for the opportunity to address this issue without having to redo all our analyses from scratch. When we started the project, the HCP pipeline data were not available for all participants and we therefore used volumetric pipelines that are standard in the field. The functional connectivity patterns between the AI and ACC obtained here are consistent with many previous neuroimaging studies. While the ICA and FIX based approaches have their strengths, they are not without limitations. Crucially, all our findings were replicated using the parcellation developed by Glasser and colleagues which is based on the multimodal surface-based pipeline described by the reviewer.

Reporting of methodological detail.There were several concerns related to the reporting of methodological detail that need to be addressed.The description of key imaging methods is inadequate. It is useful to restate the key aspects of the HCP diffusion acquisition and preprocessing for readers so they need not refer back to another paper. Additionally, the macaque data acquisition description, apparently new data acquired for the paper, is inadequate. For example, the resolution, TR/TE, b-shells, and preprocessing details are left unspecified. Of note the HCP data were acquired with a single spin echo sequence to enable minimizing the TE and maximizing the SNR. The preprocessing of the macaque diffusion data needs to be described. Further, no details are given as to how the surface-based maps of RTOP were created and any additional processing (e.g. spatial smoothing, cross-subject alignment) that was done on the HCP preprocessed diffusion data or derivatives of this. Depending on the preferred approach of the authors (ideally the one that worked the best) the RTOP data should have been computed in the individual subject's physical volume space and then mapped from between the individual's MSMAll aligned white matter and pial surface meshes onto the 32k MSMAll surface standard space without any spatial smoothing or after mapping the preprocessed diffusion data from between the individual's MSMAll aligned white matter and pial surface meshes onto the 32k MSMAll surface standard space without any spatial smoothing and then modeling RTOP. This data could then simply be averaged across subjects. With 449 subjects, smoothing is unlikely to be necessary for SNR purposes, but modest surface-based smoothing reduces precision much less than volume-based smoothing (Coalson et al., 2018 PNAS).

We thank the reviewers for this suggestion. We have addressed these concerns, by

1) Including details of the HCP dMRI acquisition protocol in the Supplementary Information Section I.B.

2) Including details of the RTOP computational procedures in the Supplementary Information Section I.C. These were done exactly as in accordance to the method envisioned by the reviewer performing a subject-space computation, projecting it to the 32k MSMAll-aligned surface mesh, specifically the mid thickness one, without any spatial smoothing and then taking advantage of the vertex correspondences across the 32k MSMAll provided surfaces to perform the whole-sample analyses.

3) Including details of the macaque dMRI acquisition and processing protocols in the Supplementary Information Section I.E. the details of the monkey dataset diffusion MRI acquisition and processing in supplementary information section I.E.

More information about the CCA analysis.The association between RTOP and cognitive control measures represents an important part of the results. While the correlation between the left and right canonical vectors is significant and features a moderate effect size, I have some issues with the interpretation of the weights in the first component of the CCA, especially because I know little about this technique.1) The right dorsal anterior insula has the highest weight, as expected and detailed in the Discussion. However it is a negative weight, while other weights, and notably the rvAI, are positive. What is the meaning of the sign and of the sign difference in the weights, in the interpretation of the microstructure-behavioural association?2) The weights of the first behavioural canonical variate appear flat, and most of them are zero. What are the implications of this for the interpretations?3) It would be useful to have a plot of the cumulative variance explained e.g. by the first N pairs of canonical variate, to assess whether a different combination of regional microstructur/behaviour adds a substatial amount of variance4) Does the significance of the correlation between the first pair of canonical variates survive if permutation testing is used instead of parametric testing?

We thank the reviewers for the opportunity to clarify the canonical correlation analysis (CCA) and our brain-behavior relationship findings. CCA is a multivariate statistical method for investigating the relationship between two sets of variables. It identifies linear combinations in both sets of variables such that they bear strong statistical associations. CCA is well suited to uncover multivariate relationships between two sets of high-dimensional features in cognitive neuroscience as it goes beyond mapping one-to-one and one-to-many features relationships. Below we address each specific comment raised by the reviewer:

1) In this study, we used functional parcellations of insula from both Deen et al.^29^ and Glasser et al.^35^ to demonstrate the reproducibility of our findings. CCA showed that the right dorsal anterior insula has the highest, but negative, weight in both insular parcellations, suggesting the robustness of the result. It suggests that RTOP of the right dorsal anterior insula (rdAI) has the highest contribution to the latent linear relationship on the brain variable set side. The negative weight of the rdAI RTOP is aligned with the functional role of the rdAI in cognitive control. Lower RTOP value may be associated with higher expression of von Economo neurons in this region, which could provide a neuronal basis for rapid signal relay between rdAI and dACC. Interestingly, both the right ventral anterior insula (rvAI) and the right posterior insula (rPI) have positive CCA weights, suggesting they have opposite contributions to the brain-behavior relationship in comparison to the rdAI. This is also consistent with weak association between their areas and cognitive control. On the behavioral variable side, results from both insular parcellation atlas show positive canonical weight of the standard scores from the ListSort, Flanker and working memory tasks and negative weights of the standard scores from the ProcSpeed and PicSeq tasks, indicating the robustness of our findings.

2) We apologize for the confusion. The near zero weights in behavioral canonical variate is due to the scale of behavioral variables. We have rescaled all the variables, conducted the CCA analysis (mean = 0, standard deviation = 1), and updated the tables. Note that the scaling does not influence estimation of canonical correlations. The CCA models and canonical correlations remain the same before and after rescaling.

3) Author response image 1 shows cumulative R square from the first to the sixth latent variables in the CCA analysis. Both parcellations from Deen et al.^29^ and Glasser et al.^35^ show similar profiles. The first set of latent variables explain more variance than the other set of latent variables, and the second set of latent variables explain the second most variance in the data.

**Author response image 1. sa2fig1:** Cumulative R square from the set of latent variables in CCA models. Functional parcellation29 and Multimodal parcellation35 show similar profiles.

4) Yes. We tested the significance of the correlation between the first pair of canonical variates using permutation tests. In each permutation, we randomly shuffled the first canonical variate in behavior data but kept the first canonical variate in RTOP data in their original order. Then, we computed the correlation coefficient between permuted data. This step was repeated 1000 times to get a null distribution of brain-behavior correlation coefficient, from which the p value of the original correlation coefficient was computed. We found that the original correlation coefficient was significant using the insular parcellation atlas from both Deen et al^29^ and Glasser et al.^35^ (*ps*<0.001).

Introduce the RTOP measure more clearly to a non-specialised audience:"Explain neuroanatomical correlates of RTOP, which is also useful to interpret the behavioural correlation with microstructure. Differences in insular microstructure are quantified using the RTOP DWI derivative. However I couldn't find an explanation of the microstructural properties that the RTOP targets in the Introduction or in the Results, with only one small mention at the end of the supplementary materials. Think e.g. about fractional anisotropy (FA): a neuroanatomist would like to know not only that there are localized differences in FA between samples of participants, but also how FA relates to the microstructure of axonal integrity and myelin sheet, in order to interpret those differences. Please include a brief description of the microstructural properties that RTOP allows to highlight, and the rationale that links these properties to the RTOP calculation. Preferably this should be present in the Introduction."

We agree with the reviewers that a better introduction to the RTOP measure enhances the quality of the manuscript. We have added, in the Introduction, an overview of the microstructural properties that RTOP captures:

“RTOP quantifies the restrictedness of water diffusion in the tissue^47^, with higher values reflecting more restricted diffusion. RTOP has been shown to be inversely correlated with the average volume of pores restricting diffusion in microbeads^47,48^. In neural tissue, spinal cord maturation has been shown to increase RTOP^49^. Recently, RTOP has been shown to be more sensitive to cellular organization than other more commonly used diffusion MRI measures, such as mean and radial diffusivity^50^. Based on these observations, we hypothesized that RTOP in grey matter would be sensitive to the cytoarchitectonic and neuronal organization of the insular cortex.”